# MTH1 protects platelet mitochondria from oxidative damage and regulates platelet function and thrombosis

Yangyang Ding[1,2,3,6], Xiang Gui[1,2,3,6], Xiang Chu[1,2,3,6], Yueyue Sun[1,2,3], Sixuan Zhang[1,2,3], Huan Tong[1,2,3], Wen Ju[1,2,3], Yue Li[4], Zengtian Sun[1,2,3], Mengdi Xu[1,2,3], Zhenyu Li[1,2,3], Robert K. Andrews[5], Elizabeth E. Gardiner [5], Lingyu Zeng [1,2,3,4] ✉, Kailin Xu [1,2,3] ✉ & Jianlin Qiao [1,2,3] ✉

Human MutT Homolog 1 (MTH1) is a nucleotide pool sanitization enzyme that hydrolyzes oxidized nucleotides to prevent their mis-incorporation into DNA under oxidative stress. Expression and functional roles of MTH1 in platelets are not known. Here, we show MTH1 expression in platelets and its deficiency impairs hemostasis and arterial/venous thrombosis in vivo. MTH1 deficiency reduced platelet aggregation, phosphatidylserine exposure and calcium mobilization induced by thrombin but not by collagen-related peptide (CRP) along with decreased mitochondrial ATP production. Thrombin but not CRP induced $Ca^{2+}$-dependent mitochondria reactive oxygen species generation. Mechanistically, MTH1 deficiency caused mitochondrial DNA oxidative damage and reduced the expression of cytochrome c oxidase 1. Furthermore, MTH1 exerts a similar role in human platelet function. Our study suggests that MTH1 exerts a protective function against oxidative stress in platelets and indicates that MTH1 could be a potential therapeutic target for the prevention of thrombotic diseases.

Although platelets do not have a nucleus, they are equipped with mitochondria which share some common features with a nucleus. They contain DNA, are surrounded by two lipid bilayer membranes and have the ability to divide during a cell cycle[1]. As the powerhouse of the cell, mitochondria play essential roles in metabolism and energy production[2]. Distinct from these processes, mitochondria also participate in the regulation of cell apoptosis[3], reactive oxygen species (ROS) generation[4] and the ER-stress response[5]. Similar to their roles in nucleated cells, mitochondria in platelets also regulate metabolism, ATP production, platelet activation and apoptosis[1,6,7], which are fundamental processes for platelet function. Therefore, impaired mitochondrial function is likely to cause platelet dysfunction and abnormal

apoptosis, which might contribute to the pathogenesis of cardiovascular disease[8], diabetes mellitus[9] and sepsis[10].

Under normal physiological conditions, the cellular DNA and nucleotide pool are continuously exposed to ROS, thereby oxidatively damaged[11,12], resulting in the generation of various types of oxidized nucleotides, such as 8-oxo-2'-deoxyguanosine-5'-triphosphate (8-oxo-dGTP), 8-oxo-2'-deoxyadenosine-5'-triphosphate (8-oxo-dATP), 2-hydroxy-2'-deoxyadenosine-5'-triphosphate (2-OH-dATP) and 2-hydroxyadenosine-5'-triphosphate (2-OH-ATP)[13–16]. During DNA replication and transcription processes, these oxidized nucleotides can be incorporated into DNA and induce base mispairing during DNA replication, causing transversions, thereby leading to genome instability

[1]Blood Diseases Institute, Xuzhou Medical University, Xuzhou, China. [2]Department of Hematology, The Affiliated Hospital of Xuzhou Medical University, Xuzhou, China. [3]Key Laboratory of Bone Marrow Stem Cell, Xuzhou, Jiangsu Province, China. [4]School of Medical Technology, Xuzhou Medical University, Xuzhou, China. [5]Division of Genome Science and Cancer, John Curtin School of Medical Research, Australian National University, Canberra, Australia. [6]These authors contributed equally: Yangyang Ding, Xiang Gui, Xiang Chu. ✉e-mail: zengly2000@163.com; lihmd@163.com; jianlin.qiao@gmail.com

and mutations[17,18]. In order to counteract oxidative damage to nucleotides or DNA, nucleotide pool sanitizing enzymes encoded by *Human MutT Homolog 1 (MTH1)*, also known as *Nudix hydrolase 1 (NUDT1)* have evolved[19,20]. MTH1 is a member of the Nudix phosphohydrolase superfamily of enzyme that hydrolyzes oxidized nucleotides, such as 8-oxo-dGTP, 2-OH-dATP and 8-oxo-dATP, into monophosphates to prevent the mis-incorporation and maintain genome stability[21,22]. MTH1 protein is mainly localized to the cytoplasm, with small portions found in the mitochondria and nucleus[23], implying that MTH1 is involved in the sanitization of mostly cytosolic-derived nucleotide pools for both nuclear and mitochondrial genomes. Due to high levels of oxidative stress in tumors, MTH1 expression is significantly increased in solid tumors, including lung[24] and colorectal cancer[25]. Additionally, a significant association of MTH1 expression with poor disease prognosis and higher relapse rates were observed in patients with pancreatic cancer or lung adenocarcinoma[26]. Inhibition of MTH1 can effectively and selectively trigger cell death in cancer cells by introducing the mis-incorporation of toxic oxidized nucleotides into DNA[27], suggesting that therapeutic targeting MTH1 might be an effective tumor-suppressive strategy for the treatment of tumors[26].

Considering that platelets contain mitochondria which play crucial roles in platelet function[1], whether MTH1 is expressed and functions in platelets remains unclear. Notably, the MTH1 transcript was detected in RNA-seq data from human platelets[28]. In the present study, we first demonstrate that MTH1 protein is expressed in the mitochondria of both human and mouse platelets. To further characterize the role of MTH1 in platelet function, we generated megakaryocyte/platelet-specific *MTH1* knockout mice and demonstrated that MTH1 deficiency selectively impaired G-protein coupled receptor (GPCR)-dependent platelet function and mitochondrial ATP production. Our study identifies a regulatory role of MTH1 in platelet function and protection of platelets against oxidative stress, implying that MTH1 might be a potential target for the treatment of thrombotic disorders.

## Results

### MTH1 is expressed in platelet mitochondria and its deficiency impairs in vivo hemostasis and thrombus formation

To assess the expression MTH1 of in platelets, we isolated platelets from healthy individuals and wild-type mice to measure MTH1 expression by western blot using different antibodies. An 18 kDa MTH1 protein was detected in both human and mouse platelet lysates by three different antibodies (Fig. 1a). To exclude the potential contamination of leukocytes, we measured the expression of CD45 (a leukocyte maker) in our preparations of isolated platelets and could not detect CD45 expression (Fig. 1a), indicating that leukocytes were below detectable levels in the prepared platelets. In addition, the specificity of one MTH1 antibody was verified in *MTH1* knockout platelets (Fig. 1b), confirming that MTH1 is expressed in platelets. A fractional immunoblotting approach was performed to further analyze the location of MTH1 in platelets and we found that MTH1 was enriched in the mitochondrial fraction (Fig. 1c). Consistently, the immune electron microscopy analysis also showed the localization of MTH1 protein to the mitochondria of platelets (Fig. 1d).

Using megakaryocyte/platelet-specific *MTH1* knockout mice (Fig. S1), we found that *MTH1* deficiency did not affect platelet count, mean platelet volume (MPV), platelet distribution width (PDW) and plateletcrit (PCT) ($P > 0.05$) (Fig. S2a). Further, it did not alter the protein or mRNA level of platelet receptors $α_{IIb}β_3$, GPIbα and GPVI ($P > 0.05$) (Fig. S2b) nor affect platelet ultrastructure or the number and size of α- and dense granules (Fig. S2c). In addition, we also found comparable platelet parameters (platelet count, MPV, PDW and PCT) in Platelet Factor 4-Cre recombinase (PF4-Cre) mice (Fig. S3).

Further analysis demonstrated that platelet-specific MTH1-deficient mice had significantly prolonged tail bleeding times (Fig. 1e) and impaired arterial thrombus formation (Fig. 1f) compared to control

mice ($P < 0.01$). In addition, we also found that MTH1 deficiency in platelets significantly inhibited venous thrombosis as shown by shortened venous thrombus length ($P < 0.01$) and reduced thrombus weight ($P < 0.001$) (Fig. 1g). To evaluate the influence of other blood cells and coagulation activity on the hemostasis and thrombosis, we measured the number of white blood cells and red blood cells as well as VWF levels and coagulation activity (the activated partial thromboplastin time) and the results showed that no significant differences in any of these parameters between control mice and MTH1-deficient mice (Fig. S4). Taken together, our data indicates that MTH1 deficiency impairs platelet hemostatic function, arterial and venous thrombus formation.

### Decreased GPCR-dependent platelet aggregation, calcium mobilization and mitochondrial ATP production in MTH1-deficient platelets

We then evaluated MTH1's effect on platelet function in vitro and showed that MTH1 deficiency did not affect CRP-mediated platelet aggregation or ATP release at low (0.1 μg/ml) (Fig. 2a) or high dose (1 μg/ml) (Fig. S5). However, MTH1 deficiency resulted in significant inhibition of platelet aggregation and ATP production in response to the low dose of thrombin (0.01 U/ml) (Fig. 2b) or U46619 (0.3 μM) (Fig. 2c), but not to a higher dose of thrombin (0.05 U/ml) (Fig. S4) or U46619 (1 μM) (Fig. S5), suggesting that MTH1 might regulate GPCR-dependent platelet function. Consistently, MTH1 deletion significantly decreased thrombin-induced integrin activation (Fig. 2d) without affecting P-selectin expression (Fig. S6). In addition, MTH1 deficiency also significantly reduced phosphatidylserine exposure (Annexin-V binding) (Fig. 2e) and impaired calcium mobilization (Fig. 2f) in platelets after stimulation by thrombin but not by CRP. To further verify whether the impaired aggregation of MTH1-deficient platelet was due to a defect in ATP release, platelets from control mice were treated with the ATP scavenger, apyrase which eliminated the difference in extent of platelet aggregation between control and MTH1-deficient mice (Fig. 2g). Moreover, the impaired MTH1-deficient platelet aggregation induced by thrombin was completely reversed by sensitizing platelets with a sub-threshold non-aggregatory concentration of ADP (1 μM), indicating that a defect in ATP release contributed to the impaired aggregation of $MTH1^{-/-}$ platelets. Since mitochondria play critical roles in platelet function through providing energy ATP, using a mitochondria-targeted deep-red fluorescence ATP probe, we observed significantly reduced mitochondrial ATP production in MTH1-deficient platelets after thrombin stimulation compared to control platelets (Fig. 2h). Taken together, our results demonstrate that MTH1 deficiency selectively inhibits GPCR-dependent platelet aggregation, phosphatidylserine exposure, calcium mobilization and mitochondrial ATP production.

### Increased oxidative damage to mitochondria DNA in thrombin-stimulated MTH1-deficient platelets

In response to oxidative stress, MTH1 hydrolyzes oxidized nucleotides to prevent their incorporation and thus maintain genome stability[29]. To assess whether MTH1 deficiency causes the abnormal oxidative damage of mitochondrial DNA (mtDNA), we measured the accumulation of 8-oxo-7′8′-dyhydro-2′-deoxyguannosine (8-oxo-dG) (originating from 8-oxo-dGTP in the nucleotide pool) in platelet mitochondria before and after stimulation. Our results showed that 8-oxo-dG accumulation was not detected in mitochondria of resting or CRP-treated $MTH1^{fl/fl}$ platelets (Figs. 3a and S7). However, 8-oxo-dG was observed after thrombin treatment. Mitochondrial 8-oxo-dG accumulation was detected in resting platelets and levels were further elevated after thrombin but not CRP treatment, compared to resting platelets. After normalization to data from resting platelets (fold-change), CRP treatment did not result in 8-oxo-dG accumulation in control or MTH1-deficient platelets, but there was significant 8-oxo-dG accumulated in mitochondria of MTH1-deficient platelets after thrombin stimulation.

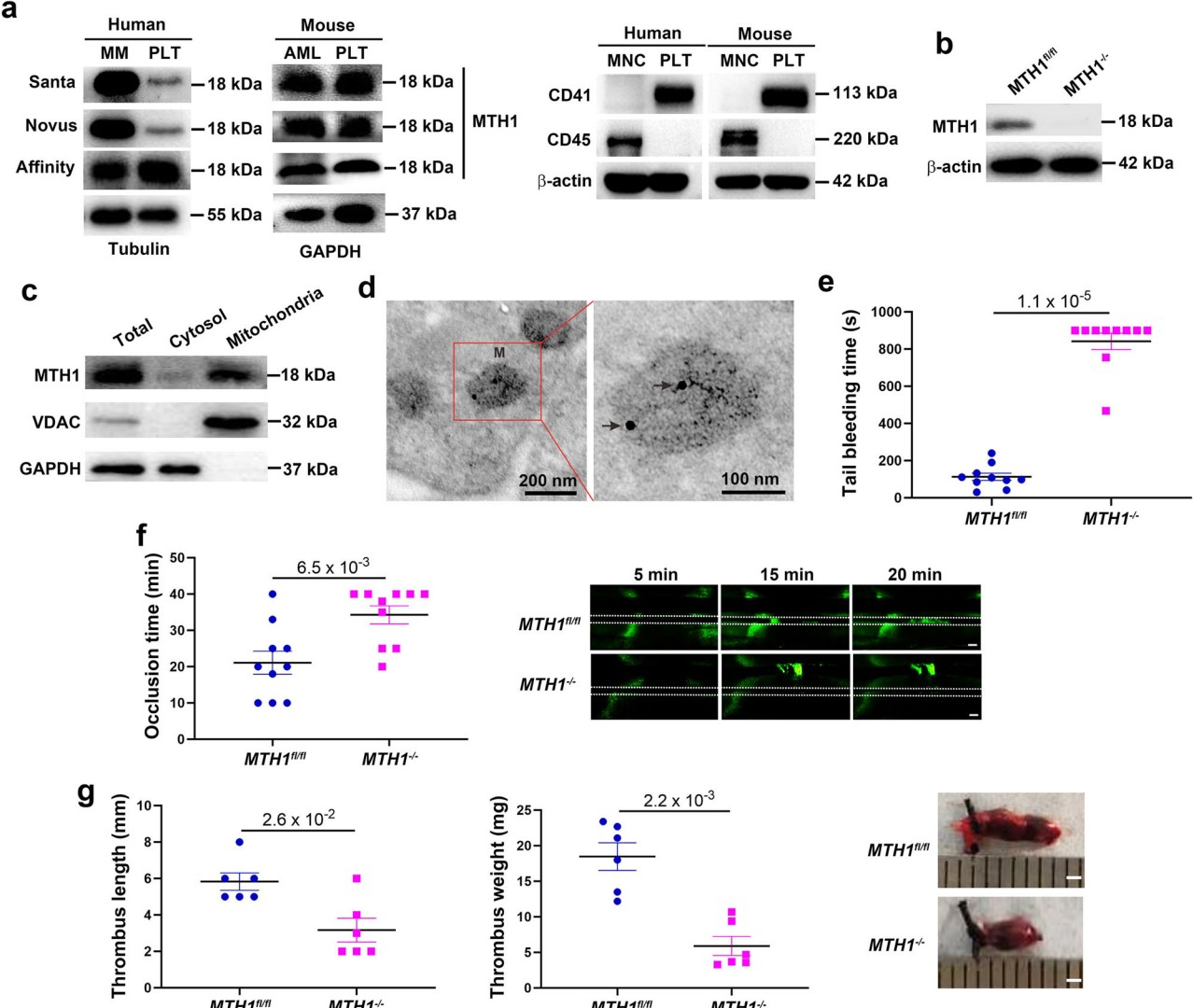

**Fig. 1 | MTH1 is expressed in platelet mitochondria and regulates hemostasis and thrombosis in vivo.** Human or wild-type mouse platelets were isolated to measure MTH1 expression by western blot with three different antibodies using human multiple myeloma (MM) cells or mouse acute myeloid leukemia (AML) cells as positive control (representative of three independent experiments) (**a**). **b** MTH1 expression in MTH1-deficient platelets was measured by western blot using MTH1 antibody (Santa Cruz). **c** Platelet cytosol and mitochondria were isolated from wild-type mice to measure the expression of MTH1 (MTH1 antibody from Santa Cruz), VDAC (VDAC antibody from Abcam) or GAPDH (representative of three independent experiments). **d** Platelets were isolated from wild-type mouse and then fixed followed by labeling with MTH1 antibody and then with the secondary antibody to evaluate the localization of MTH1 using immune electron microscopy (*n* = 3

independent isolated platelets). M indicates mitochondria and arrows indicates the positive expression of MTH1. Left panel: ×12,000 and right panel: ×30,000. **e** Tail bleeding time (mean, *n* = 10 independent animals, two-tailed Mann–Whitney test) and **f** arterial thrombus formation (mean, *n* = 10 independent animals, two-tailed Mann-Whitney test). Representative image of thrombus formation at 5, 15 and 20 min was shown. The dotted lines indicate arterial vessel walls. Scale bar = 200 μm. **g** Venous thrombosis evaluation. Mice underwent ligation of inferior vena cava (IVC) to induce venous thrombosis and IVC samples were collected after 24 h of ligation to measure thrombus length and weight (mean, *n* = 10 independent animals, two-tailed Mann–Whitney test). The representative image of the venous thrombi was shown in the right panel. Scale bar = 1 mm.

To exclude any potential false-positive signals from RNA, we performed RNase treatment before 8-oxo-dG staining and found no differences in the positive 8-oxo-dG signal in platelets treated with RNase compared to platelets treated with vehicle (Fig. S8), indicating that contamination of RNA oxidation does not contribute to the positive 8-oxo-dG signal. Analysis of 8-oxo-dG levels by ELISA also detected a significantly higher 8-oxo-dG level in thrombin-stimulated MTH1-deficient platelets compared with control platelets (Fig. S9). Since 8-oxo-dG accumulation in mitochondria is induced by oxidative stress, we measured the production of mitochondrial ROS and found a significantly reduced mitochondrial ROS in MTH1-deficient platelets compared to control platelets after thrombin stimulation (Fig. 3b), suggesting that the oxidative damage to mtDNA is mainly attributed to

MTH1 deficiency and the resultant incorporation of oxidized nucleotides rather than accentuated ROS production.

To further dissect the distinct roles of MTH1 in platelet function and mtDNA oxidative damage induced by thrombin but not CRP, we first verified that MTH1 deficient platelets had normal levels of thrombin receptors PAR3 and PAR4 (Fig. 3c). We then measured mitochondrial ROS production using immunofluorescent MitoSox Red which enables selective detection and quantification of mitochondrial superoxide. Mitochondrial ROS generation was observed after thrombin but not CRP treatment (Fig. 3d–f), suggesting that GPCR-dependent signaling can induce mitochondria ROS generation. To further verify this, platelets were pretreated with Mito-TEMPO (a mitochondria-targeted antioxidant with superoxide and alkyl radical

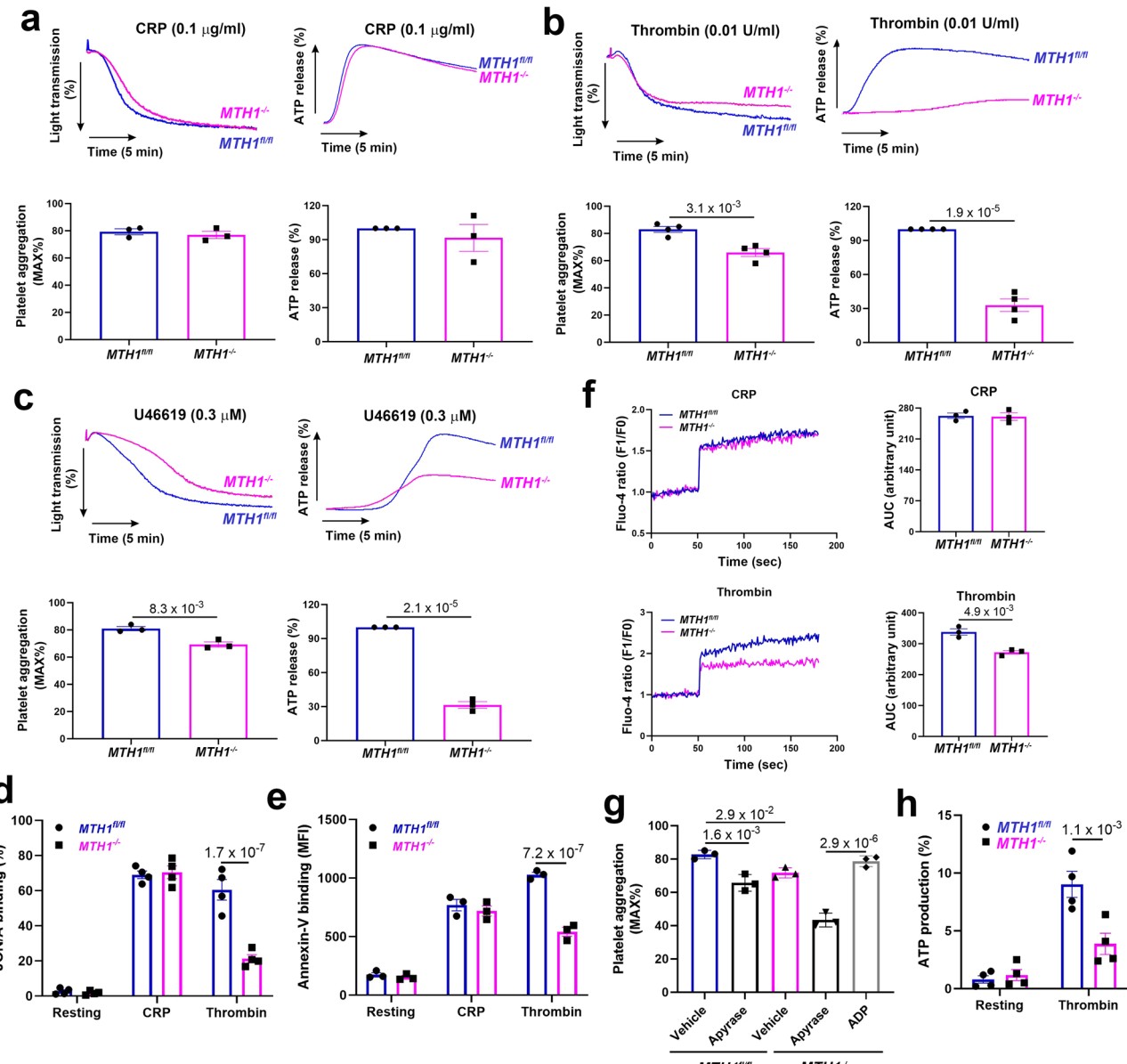

**Fig. 2 | MTH1 deficiency impairs GPCR-dependent platelet aggregation, granule secretion, calcium mobilization and mitochondrial ATP production.** Washed platelets ($200 \times 10^9$/l) from $MTH1^{fl/fl}$ or $MTH1^{-/-}$ mice were stimulated with CRP (0.1 μg/ml) (mean ± SE, $n = 3$ independent isolated platelets) (**a**), thrombin (0.01 U/ml) (mean ± SE, $n = 4$ independent isolated platelets) (**b**), U46619 (0.3 μM) (mean ± SE, $n = 3$ independent isolated platelets) (**c**) followed by analysis of platelet aggregation and ATP release (reflecting dense granule secretion) in a Lumi-Aggregometer Model 700 (two-tailed unpaired Student's $t$ test). Washed platelets were stimulated with CRP or thrombin to measure integrin αIIbβ3 activation (presented by JON/A binding) (mean ± SE, $n = 4$ independent isolated platelets, two-way ANOVA with Sidak multiple comparisons test) (**d**) and phosphatidylserine

exposure (presented by Annexin-V binding) (mean ± SE, $n = 3$ independent isolated platelets, two-way ANOVA with Sidak multiple comparisons test) (**e**) by flow cytometry or calcium mobilization using Fluo-4 AM by a microplate reader (mean ± SE, $n = 3$ independent isolated platelets, two-tailed unpaired Student's $t$ test) (**f**). **g** Platelet aggregation in response to thrombin (0.01 U/ml) after addition of apyrase (0.25 U/ml) or ADP (1 μM) (mean ± SE, $n = 3$ independent isolated platelets, one-way ANOVA with Tukey multiple comparisons test). **h** Mitochondrial ATP production was measured using the Biotracker dye in thrombin-stimulated platelets (mean ± SE, $n = 4$ independent isolated platelets, two-way ANOVA with Sidak multiple comparisons test).

scavenging properties) or apocynin (an NADPH oxidase inhibitor) and then stimulated with thrombin or CRP to measure intra-platelet ROS generation using $H_2$DCF-DA by flow cytometry[30]. As seen in Fig. 3g, Mito-TEMPO treatment significantly inhibited thrombin-induced ROS generation without affecting CRP-induced ROS production compared to vehicle treatment. However, ROS generation was significantly inhibited in both thrombin- and CRP-stimulated platelets after apocynin treatment, indicating that NADPH oxidase is involved in platelet ROS generated by treatment with either thrombin or CRP, but that mitochondria only contribute to thrombin-induced ROS production.

To evaluate which signaling pathway is involved in thrombin-induced mitochondrial ROS generation, platelets were pre-treated with different signaling pathway inhibitors. Inhibition of nuclear factor kappa B (NF-κB) (BAY 11-7082), phospholipase C (PLC) (U73122), phosphoinositide 3-kinase (PI3K) (LY294002), or Src (PP1) which are known to interfere with GPVI/CRP signaling pathways, failed to inhibit mitochondrial ROS production (Fig. 3f). However, the cell-permeant calcium inhibitor BAPTA, significantly inhibited thrombin-induced mitochondrial ROS generation, suggesting that calcium is involved in mitochondrial ROS production.

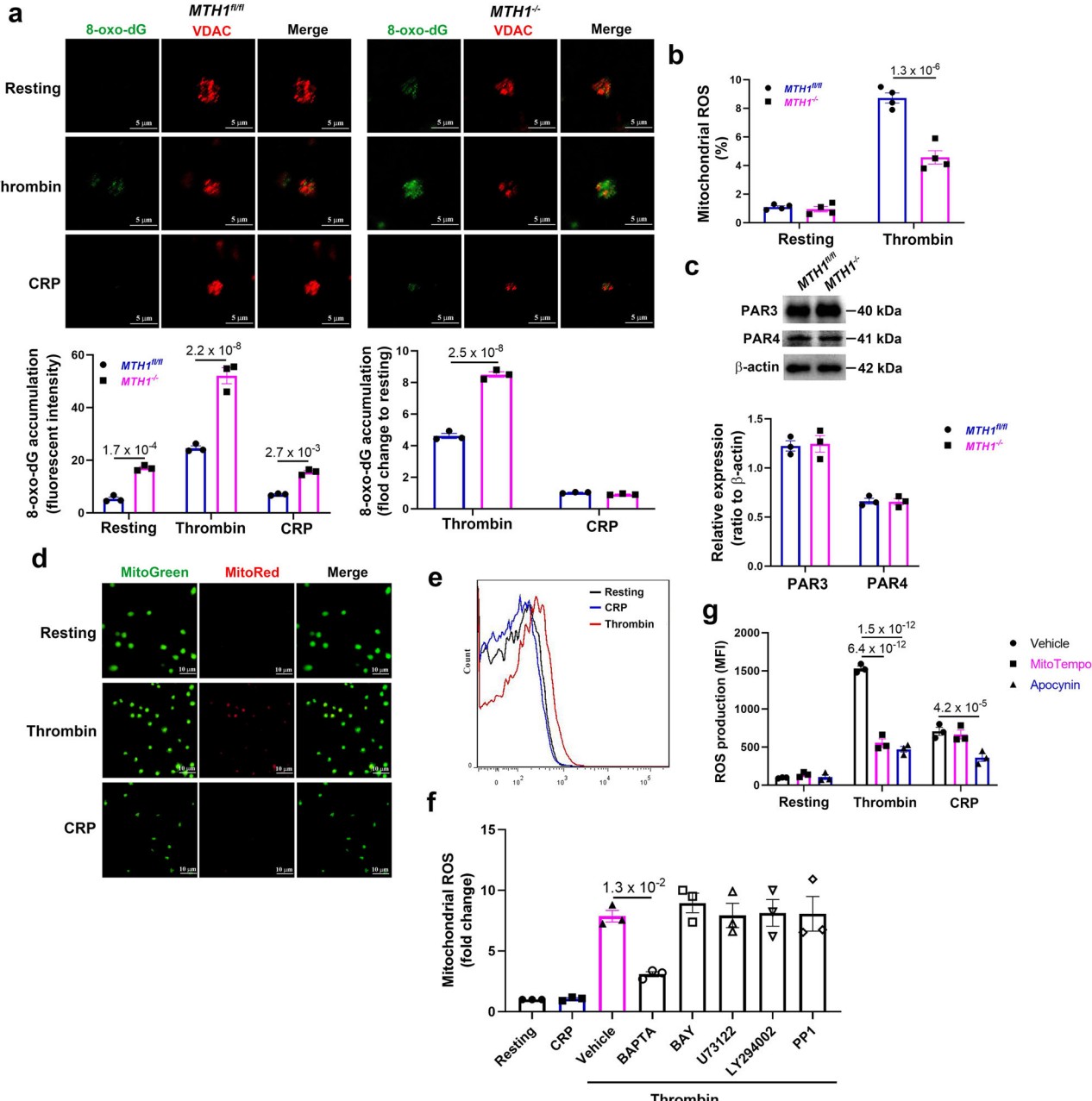

**Fig. 3 | Increased mitochondrial 8-oxo-dG accumulation in thrombin-stimulated MTH1-deficient platelets. a** Accumulation of 8-oxo-dG in mitochondria of platelets from $MTH1^{fl/fl}$ or $MTH1^{-/-}$ mice after stimulation by thrombin (1 U/ml) or CRP (5 μg/ml) (mean ± SE, n = 3 independent isolated platelets, two-way ANOVA with Sidak multiple comparisons test). **b** Washed platelets were loaded with MitoSOX Red (5 μM) for 10 min and then stimulated with thrombin (0.25 U/ml) for 3 min to measure mitochondrial ROS production by flow cytometry (mean ± SE, n = 3 independent isolated platelets, two-way ANOVA with Sidak multiple comparisons test). **c** PAR3 and PAR4 expression in $MTH1^{fl/fl}$ and $MTH1^{-/-}$ platelets under resting conditions (mean ± SE, n = 3 independent isolated platelets, two-tailed unpaired Student's t test). **d** Wild-type (WT) platelets were treated with thrombin (1 U/ml) or CRP (5 μg/ml) for 3 min followed by measuring the mitochondrial ROS generation using MitoSox Red and MitoTracker Green probes by fluorescent microscope (×100) (n = 3 independent isolated platelets). **e** Representative image of mitochondrial ROS generation in WT platelets after stimulation with CRP or thrombin by flow cytometry. **f** WT platelets were pretreated with BAPTA (calcium inhibitor) (20 μM), BAY 11-7082 (NF-κB inhibitor), U-73122 (PLC inhibitor) (5 μM), LY294002 (PI3K inhibitor) (20 μM), PP1 (Src inhibitor) (10 μM) (MedChemExpress) for 5 min followed by stimulation with thrombin or CRP to measure mitochondrial ROS by flow cytometry (mean ± SE, n = 3 independent isolated platelets, one-way ANOVA with Dunnett multiple comparisons test). **g** WT platelets were pre-incubated with vehicle, Mito-TEMPO (10 μM) or Apocynin (500 μM) and then treated with thrombin or CRP followed by measuring intracellular ROS production by flow cytometry using H2DCFDA (mean ± SE, n = 3 independent isolated platelets, two-way ANOVA with Tukey multiple comparisons test).

## Dysregulated proteins phosphorylation in MTH1-deficient platelets

To evaluate whether MTH1-deficiency alters the profile of protein phosphorylation in platelets after thrombin stimulation, we performed quantitative phosphoproteomic assays (Fig. 4a). 2580 phosphosites, 2364 phosphopeptides and 1295 phosphoproteins were identified with 2340 phosphopeptides and 2534 phosphosites being quantified in 1290 phosphoproteins (Supplemental Data 1). Based on a threshold as reported previously[31], we found 395 upregulated and 554 down-regulated phosphopeptides in $MTH1^{fl/fl}$ platelets and 774 upregulated

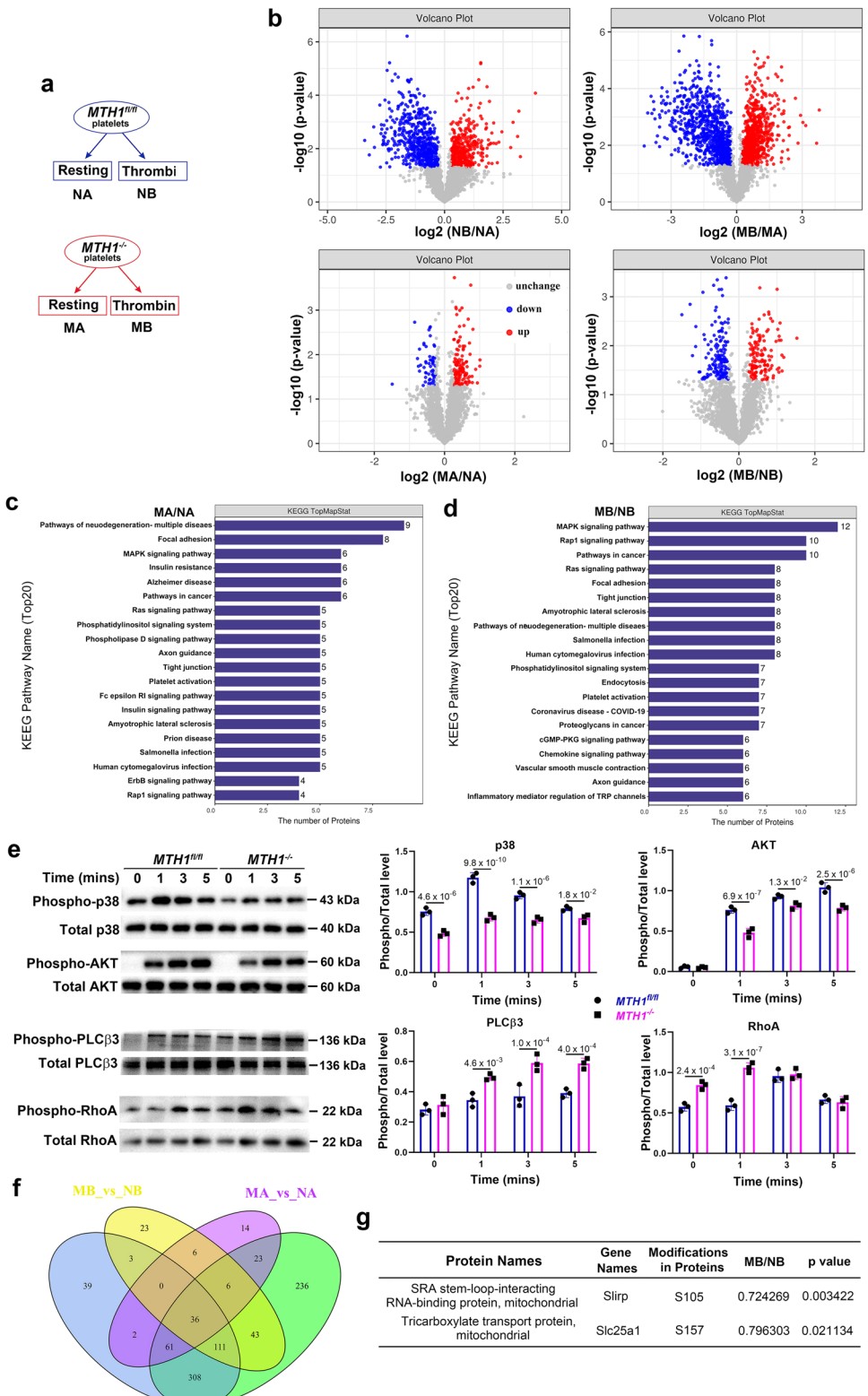

**Fig. 4 | Dysregulated protein phosphorylation in MTH1-deficient platelets after thrombin stimulation. a** MTH1$^{fl/fl}$ or MTH1$^{-/-}$ platelets were treated with thrombin (1 U/ml) for 3 min followed by quantitative phosphoproteomics assay. **b** Differentially expressed phosphopeptides between two groups were presented as volcano map. *X*-axis shows the fold change (logarithmic conversion based on 2) and *Y*-axis shows the P-value (logarithmic conversion based on 10). Red dots represented the differentially upregulated phosphopeptides with significance and Blue dots showed the differentially downregulated phosphopeptides with significance. KEGG pathway analysis between control and MTH1-deficient platelets under the condition of resting (MA/NA) (**c**) or stimulation (MB/NB) (**d**). **e** *MTH1*$^{fl/fl}$ or

*MTH1*$^{-/-}$ platelets were stimulated with thrombin (1 U/ml) followed by measuring the phosphorylation level of p38 MAPK, AKT, PLCβ3 and RhoA. The data were quantified based on three independent experiments (mean ± SD, n = 3 independent isolated platelets, two-way ANOVA with Sidak multiple comparisons test). **f** The number of differentially expressed phosphopeptides among the four groups. **g** Details of the 2 differentially expressed phosphopeptides localized in the mitochondria with significance identified from the comparison of control and MTH1-deficient platelets after thrombin stimulation (*n* = 3 independent experiments, two-tailed unpaired Student's *t* test).

and 848 downregulated phosphopeptides in MTH1-deficient platelets after stimulation. In addition, when comparing *MTH1^{fl/fl}* and *MTH1^{-/-}* platelets, we observed 123 upregulated and 55 downregulated phosphopeptides in resting platelets relative to 135 upregulated and 151 downregulated phosphopeptides in activated platelets (Fig. 4b) (Supplemental Data 2). Moreover, KEGG pathway analysis showed that these dysregulated phosphoproteins were enriched in platelet activation, MAPK signaling and phosphatidylinositol signaling pathways in MTH1-deficient platelets versus control platelets under resting condition (Fig. 4c) or thrombin stimulation (Fig. 4d). Further supporting the role of these signaling pathways, the phosphorylation of p38 MAPK and AKT were significantly decreased in thrombin-treated *MTH1^{-/-}* platelets compared to control platelets (Fig. 4e). In addition, *MTH1^{-/-}* platelets presented a significantly increased phosphorylation level of PLCβ3 (Ser1105) and RhoA (Ser188) after thrombin stimulation (Fig. 4d), indicating that MTH1 deletion impairs PLCβ3 activity and RhoA kinase signaling since PLCβ3 (Ser1105) phosphorylation inhibits its activity[32] and RhoA (Ser188) phosphorylation prevents RhoA membrane relocalization[33]. To evaluate whether MTH1 deficiency affects other GPCR signaling pathways, we evaluated phosphorylation of PLCβ3 and AKT in platelets treated with ADP or U46619. The results showed that MTH1 deficient platelets had significantly increased levels of phosphorylation of PLCβ3 at Ser1105 and decreased phosphorylation of AKT at Ser473 after stimulation with GPCR agonists, ADP or U46619 (Fig. S10), further supporting a role for MTH1 in GPCR signaling transduction. Considering the phosphoprotein overlap across different comparisons (Fig. 4f), to find the specific thrombin-mediated effects on MTH1 deficiency, we excluded phosphoprotein data obtained under the following conditions from analysis: 1) control platelets before and after stimulation; 2) MTH1-deficient and control platelets under resting condition; 3) MTH1-deficient platelets before and after stimulation. After that, 23 differentially expressed unique phosphopeptides were identified by comparison of MTH1-deficient platelets with control platelets after thrombin stimulation, including 16 downregulated and 7 upregulated (Table S1). Since MTH1 is localized to platelet mitochondria (Fig. 1d), we focused on mitochondrial phosphoproteins and found that Slirp and Slc25a1 phosphorylation was significantly downregulated in MTH1-deficient platelets (Fig. 4f). As a stem-loop RNA binding protein, Slirp has been demonstrated to regulate mitochondrial protein synthesis[34]. The reduced phosphorylation of Slirp might indicate abnormal mitochondrial protein synthesis. Slc25a1 is a tricarboxylate transport protein and mediates the exchange of mitochondrial citrate for cytosolic malate as well as the exchange of citrate for isocitrate[35]. The decreased Slc25a1 phosphorylation is in accordance with the observed impaired mitochondrial ATP generation.

## Defective expression of *cytochrome c oxidase 1* in activated MTH1-deficient platelets

Besides MTH1 as a nucleotide pool sanitizing enzyme, there are also base excision repair (BER) enzymes involving in the repair of damaged DNA, such as 8-oxoG DNA glycosylase (OGG1) and human MutY homolog (MUTYH)[36]. To evaluate whether MTH1 deficiency affects the expression of OGG1 and MUTYH, we measured their expression by western blot and found equivalent levels of OGG1 and MUTYH protein in control and *MTH1^{-/-}* platelets (Fig. 5a). In addition, MTH1 deficiency did not affect the expression of MTH2 and MTH3 in platelets (Fig. 5a) or mitochondrial biogenesis in platelets (Fig. S11) as demonstrated by no difference of mitochondrial numbers between *MTH1^{fl/fl}* mice and *MTH1^{-/-}* mice. Consistently, platelet half-life in vivo was also maintained in MTH1-deficient mice (Fig. 5b), which agrees with the normal platelet count.

Although the majority of DNA is packaged in chromosomes within the nucleus, mitochondria have their own genome containing a small amount of mtDNA with independent transcription and protein

synthesis[37]. The mitochondrial genome contains 37 genes which encode 13 proteins, 22 transfer ribonucleic acids (tRNAs) and 2 ribosomal ribonucleic acids (rRNAs)[38]. The mitochondrial gene-encoded proteins are the core and membrane-bound subunits of respiratory chain complexes I, III, IV, and V (ATP synthase) which participate in electron transport and ATP synthesis in the oxidative phosphorylation system. Considering the mtDNA oxidative damage and ATP generation induced by MTH1 deficiency and the decreased phosphorylation of Slirp (regulates mitochondrial protein synthesis), we next evaluated whether MTH1 deficiency affected the expression of proteins encoded by mtDNA. We found a significantly reduced expression of cytochrome c oxidase 1 (MT-CO1) in MTH1-deficient platelets under resting conditions and MT-CO1 expression was further reduced to an almost undetectable level after thrombin stimulation (Fig. 5c). Further, other mtDNA-encoded proteins (ND1 and ATP8) were downregulated to varying degrees. CYTB was a notable exception in MTH1-deficient platelets and its expression was unchanged. However, the expression of nuclear DNA-encoded proteins (NDUFV1, UQCRC2, COX6A1, ATP5A) remained stable in control and MTH1-deficient platelets (Fig. 5c). However, no significant changes of the expression of proteins encoded by either mitochondrial DNA (ND1 and MT-CO1) or nuclear DNA (NDUFV1 and UQCRC2) were detected in platelets stimulated with CRP (5 μg/ml) (Fig. S12), indicating that GPVI engagement and signaling does not trigger the observed oxidative damage to mitochondrion, which is consistent with the lack of mitochondrial ROS in CRP-stimulated platelets (Fig. 3d). Consistent with the reduced protein expression of MT-CO1, the expression of the *MT-CO1* gene was also decreased in MTH1-deficient platelets after thrombin stimulation (Fig. 5d). The reduced MT-CO1 expression in stimulated MTH1-deficient platelets might be due to a higher guanine content in *MT-CO1* DNA relative to other mtDNAs (Fig. 5e), making this gene more susceptible to oxidative damage during DNA replication in activated MTH1-deficient platelets. However, whilst the analysis of oxidative damage of total mtDNA was possible, it was not feasible to obtain sufficient platelet mtDNA to directly assess the oxidative modification of MT-CO1 using mtDNA immunoprecipitation approaches, which represents a limitation of the study.

Since MTH1 is both a nuclear and mitochondrial protein, its deletion in megakaryocytes might potentially cause the nuclear genome to be susceptible to oxidative damage, which could then influence the mRNA and protein profiles of platelets that are produced. To evaluate whether the changes in platelet phosphoproteomic profiles with MTH1 deletion were due solely to the effects on mitochondrial genes, cultured megakaryocytes differentiated from CD117^+ hematopoietic stem cells from *MTH1^{fl/fl}* or *MTH1^{-/-}* mice were treated with 50 μM $H_2O_2$ to induce oxidative damage. Damage to nuclear and mitochondrial DNA was assessed by quantitative PCR[39]. We found that $H_2O_2$ treatment resulted in a significantly reduced relative amplification of mtDNA in MTH1-deficient megakaryocytes relative to control megakaryocytes (Fig. S13a), indicating a more severe mtDNA damage in MTH1-deficient megakaryocytes. However, $H_2O_2$ treatment did not cause oxidative damage to nuclear DNA in both MTH1-deficient megakaryocytes and control megakaryocytes as demonstrated by no difference of the relative amplification of the nuclear β-globin gene in either experimental groups compared to vehicle-treated samples (Fig. S13b). These data indicate that mtDNA is more susceptible to oxidative damage than nuclear DNA in megakaryocytes, consistent with a previous study showing that mtDNA damage was more extensive and persisted longer than nuclear DNA damage in human cells following oxidative stress[40]. However, when comparing mitochondrial DNA damage between genotypes, there was significantly more oxidative damage to mitochondrial DNA from MTH1-deficient megakaryocytes relative to control megakaryocytes, indicating that mitochondrial DNA oxidative damage was enhanced in the setting of MTH1 deficiency. As the physiological concentration of $H_2O_2$ ranges

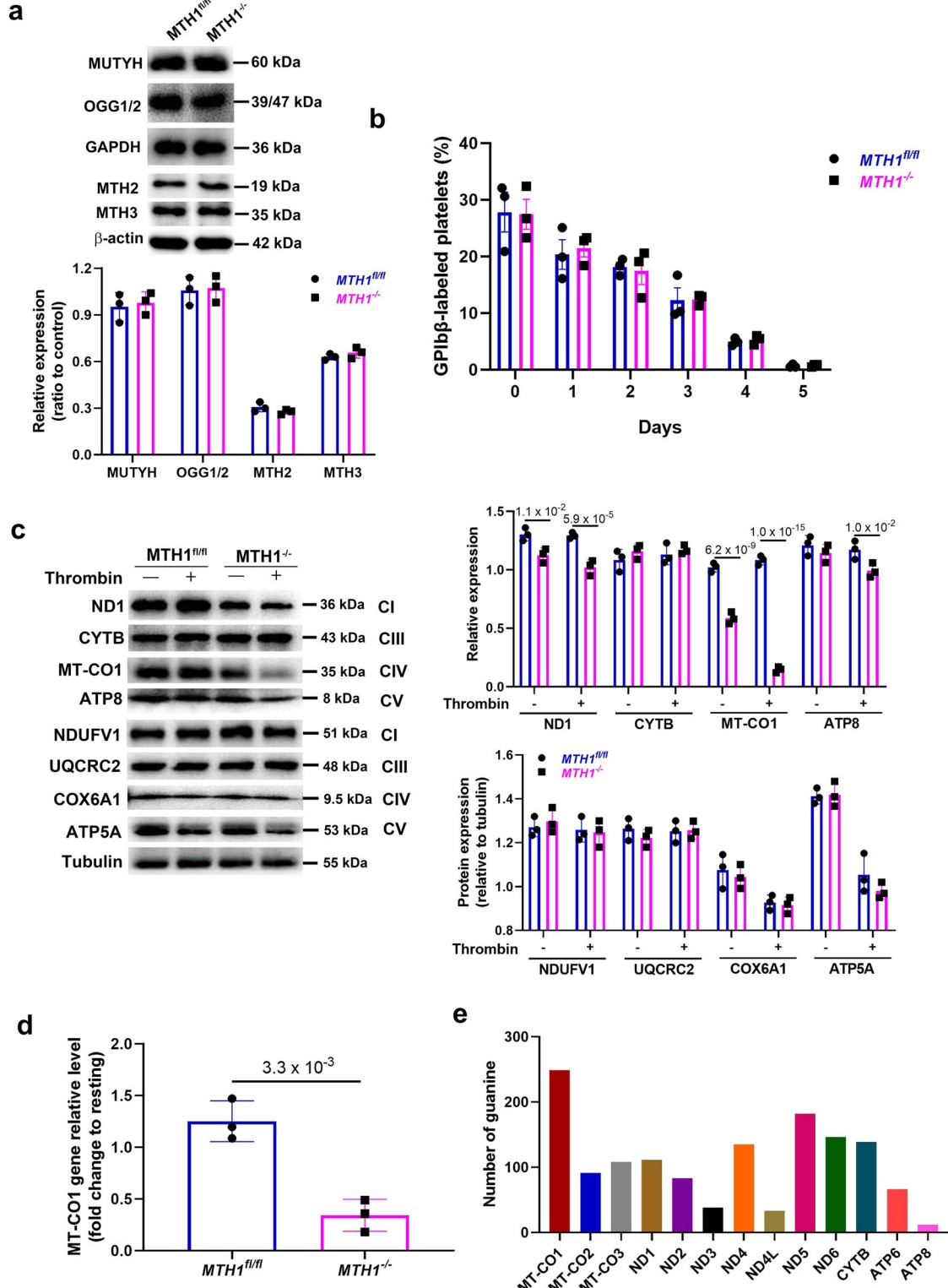

**Fig. 5 | MTH1 deficiency reduces the expression of mtDNA-encoded genes after thrombin stimulation. a** Western blot analysis of the expression of MUTYH, OGG1/2, MTH2, and MTH3 in *MTH1^fl/fl^* and *MTH1^-/-^* platelets under resting condition. The representative images were shown from three independent experiments (mean ± SD, *n* = 3 independent isolated platelets). **b** Mice were administered a Dylight 488-labeled anti-GP1bβ antibody (Emfret, X488) via tail vein (0.1 μg/g body weight) to measure platelet half-life by flow cytometry. Data were presented as mean ± SE (*n* = 3, two-way ANOVA with Sidak multiple comparisons test. **c** Expression of subunit of complexes (CI, CIII, CIV and CV) in the mitochondrial respiration chain proteins encoded by mtDNA (ND1, CYTB, MT-CO1, ATP8) or nucleus DNA (NDUFV1, UQCRC2, COX6A1, ATP5A) in platelets from *MTH1^fl/fl^* or *MTH1^-/-^* mice before (−) and after (+) stimulation with thrombin (1 U/ml) for 3 mins (representative of three independent experiments) (mean ± SD, *n* = 3, two-way ANOVA with Sidak multiple comparisons test. **d** MT-CO1 gene expression level in thrombin-activated platelets from *MTH1^fl/fl^* or *MTH1^-/-^* mice was measured by quantitative real-time PCR and represented as a fold change relative to its level in resting platelets (mean ± SD, *n* = 3 independent isolated platelets, two-tailed unpaired Student's *t* test). **e** Analysis of the number of guanine in the 13 mtDNA sequences.

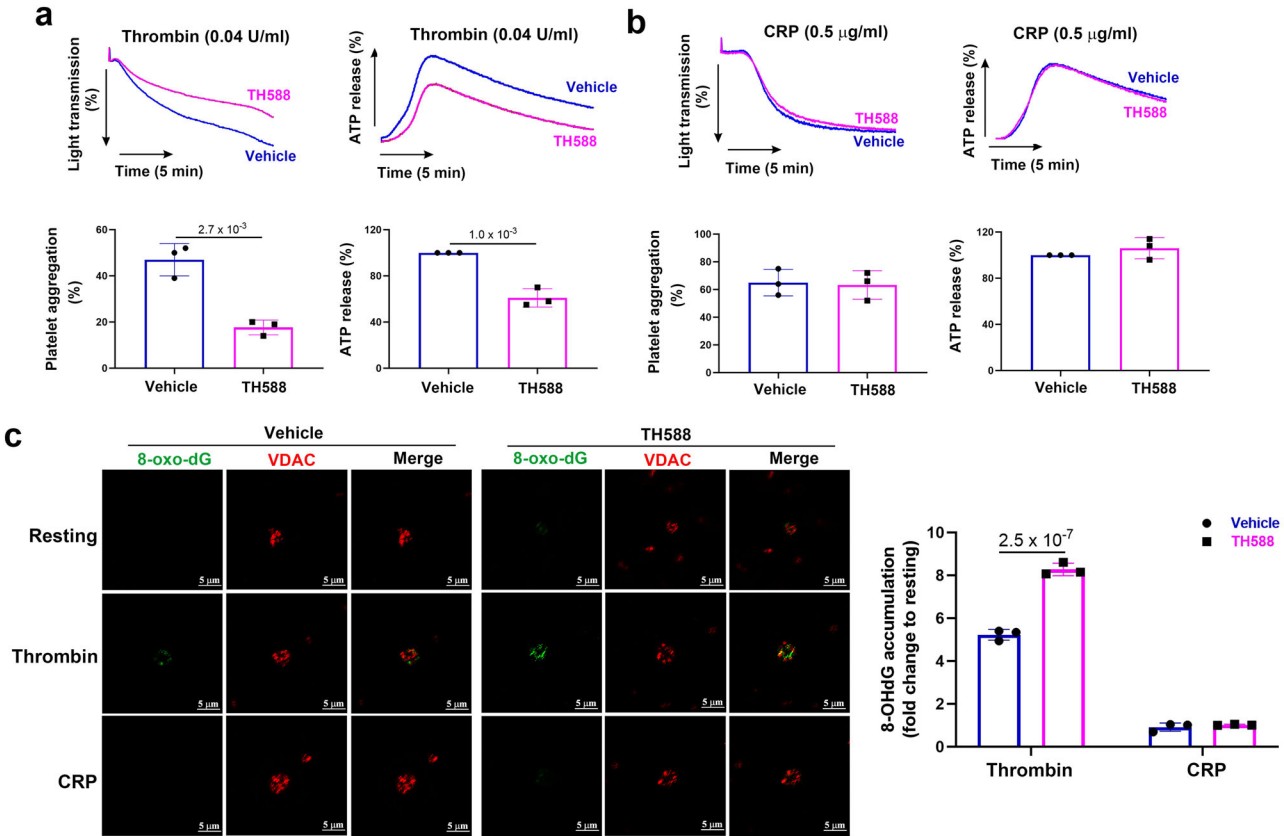

**Fig. 6 | Inhibition of human platelet MTH1 reduces thrombin-mediated platelet function.** Washed human platelets were pre-treated with 5 μM TH588 (MTH1 inhibitor) for 1 h at 37 °C and platelet aggregation and ATP release induced by either thrombin (0.04 U/ml) (mean ± SD, $n = 3$, two-tailed unpaired Student's $t$ test) (**a**) or CRP (0.5 μg/ml) (mean ± SD, $n = 3$, two-tailed unpaired Student's $t$ test) (**b**), as well as 8-oxo-dG accumulation in platelet mitochondria induced by thrombin (1 U/ml) or CRP (5 μg/ml) (**c**) (mean ± SD, $n = 3$, two-way ANOVA with Sidak multiple comparisons test) were measured.

from 1-100 nM[41], we speculated that MTH1 deletion would not result in oxidative damage to nuclear DNA in megakaryocytes under physiological condition as demonstrated by no difference of the expression of proteins encoded by megakaryocyte nuclear DNA in platelets under resting conditions (Fig. 5c). This observation is further supported by normal activity and function of $MTH1^{-/-}$ platelets after CRP stimulation. Therefore, we believe that all the changes in platelet phosphoproteomic profiles with MTH1 deletion are due solely to the effects on mitochondrial genes.

### Inhibition of MTH1 impairs human platelet function and causes mtDNA oxidative damage

Considering MTH1 deficiency significantly impairs mouse platelet function, we then investigated whether it also plays a role in human platelet function using MTH1 inhibitor TH588. Our results showed that treatment with TH-588 to inhibit MTH1 significantly reduced platelet aggregation and ATP production in response to a low (0.04 U/ml) (Fig. 6a) but not to a high dose (0.1 U/ml) (Fig. S13) of thrombin. However, TH588 did not alter platelet aggregation and ATP secretion in response to CRP (0.5 or 2 μg/ml) (Fig. 6b, Fig. S14 respectively). Moreover, MTH1 inhibition also resulted in significantly increased 8-oxo-dG accumulation in platelet mitochondria in response to thrombin but not CRP stimulation (Fig. 6c) (Fig. S15). Further, the positive 8-oxo-dG signal in human platelets did not include any RNA oxidation signal contamination as shown by no difference between positive 8-oxo-dG signal in human platelets treated with RNase versus that in platelets treated with vehicle (Fig. S16). Finally, mitochondrial ROS generation in human platelets was only observed after thrombin

stimulation (Fig. S17). Taken together, these data show that MTH1 might also play a role in human platelet function.

## Discussion

In our study, we detected MTH1 expression in the mitochondria of both human and mouse platelets and found that its deficiency in platelets selectively inhibits GPCR-dependent platelet function and mitochondrial ATP production, hemostasis and thrombus formation. Importantly, high concentration of GPCR agonists, thrombin or U46619, normalized the impaired aggregation of MTH1-deficient platelets, indicating that the high dose of agonists might stimulate the activation of alternative signaling pathways, which can bypass the mitochondrial-associated signaling and thereby overcome the platelet dysfunction associated with mitochondrial damage. Future studies will delineate the contribution of MTH1 deficiency in megakaryocytes, for example to assess differences in signaling protein expression which may also contribute to the observed platelet functional outcomes. Interestingly, MTH1 deficiency caused a profound prolongation of tail bleeding time, in contrast to the relatively small/partial reduction in the arterial thrombus formation observed. Hemostasis and thrombosis events are distinct, involving variable contributions from signaling pathways, that then differentially modulate integrin αIIbβ3 bidirectional signaling[42,43]. Selectively targeting αIIbβ3 outside-in signaling has been postulated as a therapeutic avenue to inhibit thrombosis while maintaining hemostasis in animal models[44], in line with the critical role of αIIbβ3 outside-in signaling in thrombus formation. In our study, MTH1 deficiency did not affect platelet spreading (Fig. S18) or clot retraction (Fig. S19), two processes regulated by αIIbβ3 outside-in

signaling, implying that MTH1 may not play a role in αIIbβ3 outside-in signaling, thus explaining the small or partial reduction of arterial thrombus formation in mice.

Aligned with MTH1's role in the protection of DNA or nucleotides from oxidative stress-mediated damage, MTH1 deficiency was reported to result in increased accumulation of 8-oxoG in mtDNA in the striatum followed by neuronal or terminal fibers dysfunction after administration of a neurotoxin[45,46]. In another study, MTH1-deficient fibroblasts showed a high susceptibility to oxidative stress-induced cell dysfunction and death along with 8-oxoG accumulation in both nuclear and mitochondrial DNA[47]. Consistent with published work, our study found that MTH1-deficient platelets had significantly increased oxidative damage to mtDNA after thrombin stimulation compared to control platelets as demonstrated by increased accumulation of 8-oxoG in the mitochondrial DNA of MTH1-deficient platelets. Our studies showed that thrombin but not CRP treatment can induce mitochondrial ROS generation, which contributes to the oxidative damage of mtDNA and subsequently defective thrombin-induced platelet aggregation and mitochondrial ATP production. This suggests that mitochondria might be a source of ROS generation in platelets specifically after thrombin stimulation, an idea supported by data showing significantly reduced ROS generation in thrombin-stimulated platelets after addition of Mito-TEMPO, a mitochondria-targeted antioxidant. Interestingly, a previous study showed that inhibition of mitochondrial respiration did not affect thrombin-induced ROS generation in human platelets[48]. However, in that study, ROS production was measured only after thrombin stimulation for 1 min, which may not be sufficient time to trigger thrombin-induced mitochondrial ROS generation. Supporting our hypothesis, we showed that mitochondrial ROS production was not observed in platelets within 1 min of thrombin stimulation (Fig. S20), but was found after 3 min stimulation. Moreover, our study also demonstrated that mitochondria are not the source of GPVI-stimulated ROS generation in platelets as addition of Mito-TEMPO did not affect ROS production in CRP-treated platelets. This was consistent with our previous study showing that inhibitors of mitochondrial respiration or xanthine oxidase had no inhibitory effect on CRP-induced ROS production in human platelets[30].

Since MTH1 deficiency causes mtDNA oxidative damage, we speculated that mitochondria gene expression might be affected. By measuring the expression of mtDNA-encoded proteins, we found a significantly reduced expression of MT-CO1 under resting and a further decrease to an almost undetectable level after stimulation. Considering the role of MTH1 in the hydrolysis of oxidized nucleotides (mainly 8-oxo-dGTP) together with the highest number of guanines in *MT-CO1* gene, this might enhance the susceptibility of *MT-CO1* to oxidative DNA damage relative to other mitochondrial genes, leading to its degradation and subsequently impaired protein synthesis. Leucine-rich pentatricopeptide repeat cassette (LRPPRC) is a RNA-binding protein that participates in the stabilization of most mtDNA-encoded mRNAs[49,50]. Several studies showed that LRPPRC is required for the efficient expression of the cytochrome c oxidase (COX) complex[51,52] and its mutation underpins a rare French-Canadian variant of Leigh syndrome characterized by defective expression of COX[51]. In complex with Slirp, the LRPPRC-Slirp complex regulates mt-mRNA stability, polyadenylation and mitochondrial translation[50]. In addition, Slirp has been reported to modulate mitochondrial protein synthesis and prevents LRPPRC degradation[34]. The significantly reduced Slirp phosphorylation in MTH1-deficient platelets after thrombin stimulation which was identified by quantitative phosphoproteomic analysis might indicate an impaired formation of LRPPRC-Slirp complexes, resulting in dysregulated mitochondrial translation and protein synthesis, contributing to defective expression of MT-CO1 in the MTH1-deficient platelets after stimulation by thrombin. However, whilst the phosphorylation changes within proteins such as Slirp and the changes in mitochondrial protein MT-CO1 were co-incidental, further work will be

required to demonstrate that these two events are functionally linked. Considering the significantly impaired ROS-dependent signaling pathways (reduced phosphorylation of p38 and AKT) in MTH1-deficient platelets after stimulation, it seems that MTH1 plays a dual role (pre- and post-translation) in platelet function. In addition, our data also imply that altered cytoplasmic oxidative stress indicated by impaired ROS-sensitive signaling pathways (p38 and AKT) might also directly contribute to the observed phenotype (impaired GPCR-dependent signaling) of MTH1-deficient platelets separate from the mitochondrial-related events.

In conclusion, MTH1 is expressed in the platelet mitochondria and its deficiency impairs GPCR-dependent platelet function and thrombus formation, indicating that MTH1 might be a potential target for preventing thrombotic defects.

## Methods

### Animals
Mice aged 6–10 weeks old with an equal sex ratio were used in this study. All mice were housed, bred and maintained in the Laboratory Animal Center under standard husbandry conditions at Xuzhou Medical University under 12 h/12 h light-dark cycles, controlled temperatures (22–24 °C) and 40-50% humidity with free access to food and water. C57BL/6-Tg (Pf4-icre) Q3Rsko/J mice (Strain #: 008535) were purchased from Jackson Laboratory. C57BL/6J wide-type mice were purchased from Beijing Vital River Laboratory Animal Technology Co., Ltd. The control mice were *MTH1* floxed and negative Cre recombinase with matched genetic background, age and sex. Both male and female mice were used in this study and selected using a randomized approach throughout the study. To minimize animal suffering, all possible efforts were made and mice were euthanized by $CO_2$ inhalation. All mice were fed on a normal chow diet (#P1101F, Shanghai Pluteng Biotechnology Co., Ltd., China). The experimental procedures were approved by the Animal Care and Use Committee of Xuzhou Medical University and performed in accordance the guide for the Care and Use of Laboratory Animals published by the U.S. National Institutes of Health.

### Generation of megakaryocyte/platelet-specific *MTH1* knockout mice
Megakaryocyte/platelet-specific *MTH1* knockout mice (Strain #: CKOCMP-17766-Nudt1-B6J-VA) was generated in Cyagen Biosciences Inc. (Suzhou, China). The *MTH1* gene (NCBI Reference Sequence: NM_008637.1) is located on mouse chromosome 5 with 5 exons. Exon 4 was selected as the conditional knockout (cKO) region and deletion of this region resulted in the loss of function of MTH1 (Fig. S1). To engineer the targeting vector, homologous arms and the cKO region was amplified by polymerase chain reaction using bacterial artificial chromosome clone RP23-122A5 and RP23-267D12 from the C57BL/6J library as template. For the targeting vector, the Neo cassette was flanked by self-deletion anchor sites (Fig. S1). Diphtheria toxin A was used for negative selection. C57BL/6 ES cells were used for gene targeting. The megakaryocyte/platelet-specific *MTH1* knockout mice were generated by crossing floxed mice bearing cKO *MTH1* with transgenic mice expressing *PF4* promoter-driven Cre recombinase (The Jackson Laboratory).

### Human ethics
All experimental procedures involving collection of human blood were approved by the Human Ethic Committee of Xuzhou Medical University. Written informed consent was obtained from all human volunteers (blood donors). Findings apply to male and female. Sex and gender data were not collected because the purpose of human platelet study is to evaluate the effect of the inhibition of MTH1 on platelet function through comparison of vehicle treatment with MTH1 inhibitor treatment rather than the comparison between different

individuals. Human volunteers in good health, aged 20-50 years old without taking any medications during the sample collection. Healthy volunteers were randomly recruited via flyer advertisements without any self-selection by the investigators. Study participants providing blood donations specifically for this research received a small financial compensation for their time, effort, and discomfort associated with the donation process.

## Platelet preparation
Mouse blood was drawn into tubes anticoagulated with trisodium citrate, glucose and citric acid (ACD) and then centrifuged to isolate platelets. For preparation of human platelets, ACD-anti-coagulated venous blood was centrifuged at $120 \times g$ for 20 min to obtain platelet-rich plasma (PRP) which was centrifuged at $1350 \times g$ for 15 min, washed and resuspended in Tyrode's buffer. The isolated platelets were allowed to rest for 1 h at room temperature before use. For the recruitment of healthy volunteers, sex and gender were not collected and they were randomly recruited without any self-selection by the investigators.

## Platelet receptors expression
Integrin αIIbβ3 expression was measured using FITC Rat Anti-Mouse CD41 antibody by flow cytometry (BD Pharmingen)[31,53,54]. The mRNA expression of GPIbα and GPVI was detected by quantitative real-time PCR using primers for GPIbα: Forward primer: 5′-AGTTCA-TACTACCCACTGGAGCC-3′, Reverse primer: 5′-GTGGGTTTATGAG TTGGAGGC-3′; GPVI: Forward primer: 5′-AGGAGACCTTCCATCT TACCCA-3′, Reverse primer: 5′ GAGCAAAACCAAATGGAGGG-3′.

## Ultrastructure analysis
Platelet ultrastructure and mitochondria were assessed using a transmission electron microscope. Mouse platelets were fixed in 3% glutaraldehyde, dehydrated and embedded to obtain an ultrathin section using LKB-V ultramicrotome followed by being stained with lead citrate and uranyl acetate. Ultrastructure was evaluated under a transmission electron microscope (JEOL-1200EX) and images were obtained using a Morada G2 digital camera.

## Immuno-electron microscopy
Platelets were isolated from wild-type mice and fixed followed by dehydration, resin penetration, embedding and polymerization. The resin blocks were cut into sections with 70–80 nm thin and fished out onto the 150 meshes nickel grids with formvar film. After blocking with 1% BSA in TBS solution (G0015-500ML, Servicebio), the sections were incubated with MTH1 antibody (sc-271082, Santa Cruz, 1:20) and then with a secondary antibody conjugated with 10-nm gold particles. The sections were stained with 2% uranium acetate saturated alcohol solution and dried followed by observation under a HITACHI transmission electron microscope (HT7800/HT7700).

## Platelet aggregation and ATP secretion
Platelet aggregation was performed in a Lumi-Aggregometer Model 700 (Chrono-log Corporation, Havertown, PA, USA) at 37 °C with stirring (1000 rpm) before and after stimulation. The ATP secretion was monitored in parallel with platelet aggregation after adding luciferin/luciferase reagent (Chrono-log Corporation) to platelet suspensions. ATP secretion was quantified relative to control mouse platelets.

## Platelet alpha-granule secretion and αIIbβ3 activation
Platelet alpha-granule release (degranulation) was assessed by measuring surface expression of the α-granule glycoprotein, P-selectin by flow cytometry using PE-conjugated anti-P-selectin antibody (Ebioscience, 12-0626-82, 1:20). αIIbβ3 activation was evaluated using PE-conjugated JON/A antibody (Emfret, M023-2, 1:20)[31,53]. FITC-

conjugated anti-CD41a antibody (BD Bioscience, 553847, 1:20) was used to set the platelet gate in the Forward Scatter and Side Scatter.

## Detection of VWF level
Plasma was isolated from mice and VWF levels were measured by a commercial ELISA kit (Mouse von Willebrand Factor ELISA Kit, CSB-E08439m, CUSABIO) according to the manufacturer's instructions.

## Measurement of coagulation time
Plasma was extracted from mice and the activated partial thromboplastin time was detected on an automated coagulation analyzer (Sysmex CS-5100).

## Tail bleeding assay, arterial and venous thrombus formation
A 10-mm segment of tail tip was cut off and immersed in pre-warmed sterile saline solution (37 °C) followed by measuring tail bleeding time. The bleeding time was defined as the time at which bleeding had visibly ceased[31,53].

For analysis of arterial thrombosis, platelets ($1 \times 10^8$) were labeled with calcein-AM (Santa Cruz, sc-203865) and injected into mice via tail vein injection. Following anesthetic administration, the mesenteric arterioles were surgically exposed and treated with tissue soaked in 10% w/v FeCl₃ solution to induce thrombus formation which was dynamically observed by fluorescence microscopy (Olympus IX53). The occlusion time (time to occlusion) was defined as the time from FeCl₃ application to the cessation of blood flow in the vessel as defined by the flow probe.

Venous thrombus formation was induced through ligation of the inferior vena cava (IVC) with a 2-0 non-absorbable suture using a stenosis method[31,53]. After 24 h of ligation, mice were humanely sacrificed and the abdominal cavity was surgically opened. The IVC and associated thrombi were removed and the thrombus length and weight were measured.

## Phosphatidylserine exposure
Phosphatidylserine exposure was quantified by measuring Annexin-V binding by flow cytometry. In brief, mouse platelets were stimulated with CRP (5 µg/ml) or thrombin (0.5 U/ml) for 30 min and then stained with FITC-Annexin-V (Biolegend, 640906, 1:20) for 15 min followed by detecting Annexin-V binding by flow cytometry. Annexin-V binding was presented as a mean fluorescence intensity (MFI).

## Calcium mobilization
Mouse platelets were preloaded with Fluo-4 AM (5 µM, Yeason, 40704ES50) for 30 min in Tyrode's buffer without calcium and then left for another 30 min. After washing, platelets were stimulated with CRP (0.5 µg/ml) or thrombin (0.005 U/ml) in the presence of 2 mM CaCl₂ and the fluorescence intensity was detected using a microplate reader (Biotek Synergy H1) at Ex/Em = 490/525 nm. Calcium mobilization was quantified as the ratio of the fluorescence intensity after stimulation relative to that resting platelets.

## Platelet half-life measurement
Murine platelet half-life in vivo was evaluated through administration of Dylight 488-labeled anti-GP1bβ antibody (Emfret, X488) via tail vein (0.1 µg/g body weight). The proportion of fluorescently-labeled platelets at different time points after administration was monitored by flow cytometry.

## Isolation of mitochondria and mitochondria DNA
Mitochondrial and cytosolic compartments were extracted from platelets using the Mitochondria Isolation Kit (Beyotime Biotech Inc, C3601) according to the manufacturer's instructions. Briefly, platelets (free of contamination by leukocytes as shown by undetectable CD45) were incubated with mitochondrial lysis buffer for 10 min on ice and

then homogenized followed by centrifugation at $1000 \times g$ for 10 min at 4 °C and subsequent re-centrifugation of the supernatant at $11,000 \times g$ for 10 min at 4 °C. The pellet containing mitochondria was resuspended to obtain the mitochondria and the supernatant was centrifuged at $12,000 \times g$ for 10 min at 4 °C to obtain the cytosol.

The mtDNA was isolated from platelets using Mitochondrial DNA Extraction Kit in accordance with the kit instructions (Scientific Phygene, PH1592).

## Quantitative real-time PCR

The gene expression of *MT-CO1* was quantified by quantitative real-time PCR on LightCycler® R480 II (Roche Life Science). The primer sequence was: Forward-5′-TCAACATGAAACCCCCAGCCA-3′ and Reverse-5′-GCGGCTAGCACTGGTAGTGA-3′ for *MT-CO1* and Forward-5′-TCCTTCATGTCGGACGAGGC-3′ and Reverse-5′-AATGCTGTGGC-TATGACTGCG-3′ for *CYTB*. *CYTB* was used as an internal control and the gene expression of *MT-CO1* was calculated using the $2^{-\Delta\Delta Ct}$ method relative to control.

## Immunoblotting

Immunoblotting assays were performed using antibodies against MTH1 (Santa Cruz (sc-271082, 1:400), Novus Biologicals (NB100-109SS, 1:1000), Affinity Biosciences (DF7359, 1:000)); MUTYH (sc-374571, 1:400), ATP5A (sc-136178, 1:400), UQCRC2 (sc-390378, 1:400) and OGG1/2 (sc-376935, 1:400) (Santa Cruz); MTH2 (Signalway Antibody LLC, 31526, 1:1000); MTH3 (Bioss Antibodies, bs-9514R, 1:1000); ND1 (19703-1-AP, 1:1000), CYTB (55090-1-AP, 1:1000), ATP8 (26723-1-AP, 1:1000), COX6A1 (11460-1-AP, 1:1000), and NDUFV1 (11238-1-AP) (Proteintech, 1:1000); MT-CO1 (ABclonal, A17889, 1:1000); p38 MAPK (anti-Thr180/Tyr182, Cell Signaling Technology, 8690, 1:1000); AKT (anti-Ser473, Cell Signaling Technology, 9271, 1:1000; pan-AKT, Affinity Biosciences, AF6261, 1:1000)); PLCβ3 (anti-Ser1105 (bs-3341R), Bioss Antibodies, 1:1000; pan-PLCβ3 (AF4754), Affinity Biosciences, 1:1000)); RhoA (anti-Ser188 (AF3352, 1:1000) and pan-RhoA (AF6352, 1:1000), Affinity Biosciences). The internal controls were: GAPDH (Bioworld, BS72410, 1:1000), β-actin (Bioworld, BS1002, 1:1000) or tubulin (Ab-mart, M30109, 1:1000).

## Intracellular ROS detection

Platelet intracellular ROS was measured using 2′,7′-dichlorofluorescein (H2DCF-DA) (MedChemExpress, HY-D0940). Washed mouse platelets were incubated with H2DCF-DA (10 μM) at 37 °C for 30 min and then stimulated with thrombin (0.25 U/ml) or CRP (2 μg/ml) to measure ROS generation by flow cytometry. For some experiments, platelets were pretreated with the mitochondria-targeted antioxidant Mito-TEMPO (10 μM) (MedChemExpress, HY-112879) or Apocynin (500 μM) (NADPH oxidase inhibitor) (Sigma-Aldrich, 178385) prior to stimulation by thrombin or CRP.

## Measurement of 8-oxo-dG

Washed mouse platelets ($3.0 \times 10^{8}$/ml) were stimulated with thrombin (1 U/ml) for 3 min followed by measuring 8-oxo-dG level using Mouse 8-OHdG ELISA Kit (WELL BIO, EM3698M) in accordance with manufacturer's instructions.

## Quantitative phosphoproteomics analysis

Protein was extracted from platelets in a lysis buffer containing SDT (4% [w/v] SDS, 100 mM Tris-HCl, 1 mM DTT, pH7.6) and quantified using a BCA Protein Assay Kit (Bio-Rad, USA). Equal amounts of protein were electrophoresed on 12.5% SDS-PAGE gel. The protein bands were visualized by Coomassie Blue R-250 staining followed by filter-aided sample preparation using 200 μg of proteins. Peptide mixture (100 μg) were labeled using a tandem mass tag (TMT) (Thermo Fisher Scientific) or isobaric tags for relative and absolute quantitation (iTRAQ) (Applied Biosystems) reagent. Phosphopeptides enrichment (IMAC enrichment method) was performed using High-SelectTM Fe-NTA Phosphopeptides Enrichment Kit (Thermo Scientific). After lyophilization, the phosphopeptides were resuspended in 20 μl loading buffer (0.1% [v/v] formic acid).

LC-MS/MS analysis was performed on a Q Exactive HF mass spectrometer (Thermo Fisher Scientific) coupled to Easy nLC (Thermo Fisher Scientific) for 120 min (Shanghai Applied Protein Technology Co., Ltd, Shanghai, China). After loading of peptides onto a reverse phase trap column (Thermo Scientific Acclaim PepMap100, 100 μm × 2 cm, nano-Viper C18) connected to the C18-reversed phase analytical column (Thermo Scientific Easy Column, 10 cm long, 75 μm inner diameter, 3 μm resin) in buffer A (0.1% Formic acid), they were separated in a linear gradient of buffer B (84% acetonitrile and 0.1% Formic acid) at 300 nl/min (flow rate). A positive ion mode was selected for MS analysis and the data was obtained using a data-dependent top10 method through selecting the most abundant precursor ions from the survey scan (300–1800 $m/z$) for HCD fragmentation. Automatic gain control (AGC) target was set to 3e6 with the maximum inject time of 10 ms. Dynamic exclusion duration was 40.0 s. Survey scans were acquired at a resolution of 70,000 at $m/z$ 200 and resolution of 17,500 at $m/z$ 200 was chosen for HCD spectra with an isolation width of 2 $m/z$. MS/MS spectra were searched using MASCOT engine (Matrix Science, London, UK; version 2.2) embedded into Proteome Discoverer 2.4 for identification and quantitation of phosphorylated proteins.

For bioinformatic analysis, hierarchical clustering was analyzed using Cluster 3.0 (http://bonsai.hgc.jp/~mdehoon/software/cluster/software.htm) and Java Treeview software (http://jtreeview.sourceforge.net). Motifs analysis were conducted by MeMe (http://meme-suite.org/index.htm) after extraction of the amino acid sequences containing the modified site and six upstream/downstream amino acids from the modified site (13 amino acid sites in total). Subcellular localization was analyzed using CELLO (http://cello.life.nctu.edu.tw/), a multi-class SVM classification system. After finding homolog sequences using NCBI BLAST+ client software and InterProScan, gene ontology (GO) terms were mapped and sequences were annotated using Blast2GO software. After annotation, the interested proteins were blasted against the online Kyoto Encyclopedia of Genes and Genomes (KEGG) database (http://geneontology.org/) to retrieve KEGG orthology identifications and mapped to pathways in KEGG.

## Immunofluorescence staining

Platelets were fixed in 1% paraformaldehyde and treated with 0.2% Triton followed by incubation with antibodies against VDAC (Abcam, ab15895, 1:200), 8-hydroxy-2′-deoxyguanosine (8-oxo-dG) (JaICA, N45.1, 1:50) or isotype control antibody IgG (Beyotime Biotech Inc, A7028, 1:100) and then with subsequent fluorescence-labeled secondary antibodies. Then, platelets were mounted with Vectashield and bound antibody was visualized using a confocal microscope (Zeiss LSM880). The results were quantified as the relative fluorescence intensities using Image J software.

## Mitochondrial ROS and ATP measurement

Platelets ($15 \times 10^{6}$/ml) were loaded with the fluorogenic probe MitoSOX Red (5 μM, Yeasen, 40778ES50) and MitoTracker Green (100 nM, Yeasen, 40742ES50) for 10 min and then stimulated with thrombin (1 U/ml) or CRP (5 μg/ml) followed by observation under an immunofluorescence microscope. Mitochondrial ATP production was detected using BioTracker ATP-Red Live Cell Dye (5 μM) (Sigma-Aldrich, SCT045) by flow cytometry using Flowjo (V10.07) software.

## Megakaryocyte culture, differentiation, ploidy and platelet formation

Mouse bone marrow was flushed from femurs and tibia to obtain bone marrow cells followed by isolation of CD117+ hematopoietic stem cells (HSCs) using CD117 beads (Miltenyi Biotec). The CD117+ HSCs were

cultured in Stemspan medium (STEMCELL Technologies) in the presence of 10 µg/ml LDL, 10 ng/ml IL-3, 10 ng/ml IL-6, 40 ng/ml mSCF, 50 ng/ml TPO (STEMCELL Technologies) and 1% penicillin-streptomycin (VICMED) to induce HSCs differentiation. On day 3, megakaryocytes were collected using a discontinuous density BSA gradient method.

### DNA oxidative damage

After treatment with $H_2O_2$ (50 µM) for 30 min at 37 °C to induce oxidative damage, megakaryocytes DNA was isolated for QPCR amplification of the genome gene (β-globin gene) and mitochondrial fragmentation to assess the DNA oxidative damage[39]. The primer sequences for β-globin: Forward-5′-TTGAGACTGTGATTGGCAAT GCCT-3′ and Reverse-5′-CCTTTAATGCCCATCCCGGACT-3′; DNA polymerase gene β: Forward-5′-TATCTCTCTTCCTCTTCACTTCT CCCCTGG-3′ and Reverse-5′-CGTGATGCCGCCGTTGAGGGTCTCCTG-3′; 10-kb mitochondria fragment: Forward-5′-GCCAGCCTGACCCA-TAGCCATATTAT-3′ and Reverse-5′-GAGAGATTTTATGGGTGTATTG CGG-3′; 117-bp mitochondria fragment: Forward-5′-CCCAGCTACTAC-CATCATTCAAGT-3′ and Reverse-5′-GATGGTTTGGGAGATTGGTTG ATG-3′. The relative amplification was defined as a ratio relative to the vehicle-alone treatment.

### Statistical analysis

Data are shown as mean ± standard deviation (SD) or standard error (SE) where indicated. GraphPad Prism 8.0.2 (GraphPad software, Inc) was used for statistical analysis and data normality was evaluated by the Shapiro–Wilk test. Normally distributed data were analyzed using Student's $t$ test, one-way or two-way ANOVA where indicated in the figure legend. Non-normally distributed data were assessed by the Mann–Whitney $U$ test. $P < 0.05$ indicates a significant difference.

### Reporting summary

Further information on research design is available in the Nature Portfolio Reporting Summary linked to this article.

## Data availability

The mass spectrometry proteomics data have been deposited to the ProteomeXchange Consortium (http://proteomecentral. proteomexchange.org) via the iProX partner repository[55,56] with the dataset identifier PXD041528. The authors declare that the necessary data required to validate the findings of the paper can be found within the article itself, in the Supplementary Information, or Source data file. Source data are provided with this paper.

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

## Acknowledgements

This work was supported by National Natural Science Foundation of China (grant nos. 82322005, 82261138554, 82170130, 81970124, and 81400082) [J.Q.], the Natural Science Foundation of Jiangsu Province (grant no. BK20140219) [J.Q.], the funding for the Distinguished Professorship Program of Jiangsu Province, the Shuangchuang Project of Jiangsu Province, the 333 projects of Jiangsu Province (BRA2017542) [J.Q.], the Natural Science Foundation of the Jiangsu Higher Education Institutions of China (18KJA320010) [J.Q.], Jiangsu Province's Graduate Scientific Research Innovation Program (KYCX22-2896 [X.C.] and KYCX21-2691 [X.G.]), Youth Science and Technology Innovation Team of Xuzhou Medical University and the National Health and Medical Research Council, Australia. We thank Professor Steve P. Watson (University of Birmingham, UK) for critically reading the manuscript and providing valuable comments.

## Author contributions

Y.D., X.G., and X.C. performed research, analyzed data and wrote the manuscript. Y.S., H.T., S.Z., W.J., Y.L., Z.S., M.X., and Z.L. performed research and analyzed data, E.G. and R.A. provided the intellectual input, L.Z., K.X., and J.Q. conceived and designed the study and wrote the manuscript.

## Competing interests

The authors declare no competing interests.
