## [Peer Review File · Nature Communications]

REVIEWER COMMENTS

Reviewer #1 (Remarks to the Author):

Ding et al aimed to investigate the role of the nucleotide pool sanitization enzyme, human MutT Homolog 1 (MTH1), in platelet function and thrombosis. The study found that MTH1 is present in both human and mouse platelets, and deficiency of MTH1 in platelets/megakaryocytes prolonged tail-bleeding time and impaired both arterial and venous thrombus formation. The study also found that MTH1 deficiency significantly reduced platelet aggregation, phosphatidylserine exposure, and calcium mobilization induced by thrombin but not by collagen-related peptide (CRP). Further, MTH1 deficiency impaired platelet metabolism after thrombin stimulation and caused oxidative damage to platelet mitochondrial DNA, reducing the expression of cytochrome c oxidase 1 (MT-CO1) in thrombin-stimulated platelets. Overall, the study provides novel insights into the regulatory role of MTH1 in platelet function and thrombus formation, and supports the potential of MTH1 as a therapeutic target for prevention of thrombotic or cardiovascular diseases.

However, there are several issues that need to be addressed to improve the manuscript:

Major comments

1. English writing is a little bit poor, especially many scientific terminology is wrong, so need to be proofread by English native scientists.
2. Terminology: 8-OH-dG can be used only for name of Kit in Material and Methods. In main text, it should be 8-oxo-7'8'-dihydro-2'-deoxyguanosine (8-oxo-2'-deoxyguanosine or 8-oxo-dG). Please define 8-oxo-dGTP, 2-OH-dATP, and 8-oxo-dATP, in the Introduction with their full-names and abbreviations, with appropriate references.
3. Terminology: page 10, line 17-18: "such as 8-OHdG DNA glycosylase (OGG1)" is wrong, it should be "8-oxoG DNA glycosylase (OGG1)" which excises the oxidized base 8-oxoG, not 8-oxo-dG.
4. Introduction, page 3, line 4; "a double plasma membrane" may not be appropriate for membranes of mitochondria and nucleus, "two lipid bilayer membranes" may be better.
5. Introduction, page 3, line 14-15: "Under normal physiological conditions, the cellular DNA and nucleotide pool are continuously exposed to ROS-induced oxidative damage (11, 12)" may be "Under normal physiological conditions, the cellular DNA and nucleotide pool are continuously exposed to ROS, thereby oxidatively damaged (11, 12)".
6. Introduction, Page 3, line 20: "repairing enzymes such as Human MutT homolog 1(MTH1)" is wrong, because MTH1 is not a repairing enzyme, it is "a nucleotide pool sanitizing enzyme".

7 . In Figure 1, it would be important to provide platelet-specific markers to demonstrate the purity of platelets for MTH1 quantification. This would help to exclude the possibility of contamination by other cell types that could affect the interpretation of the results. To demonstrate mitochondrial localization of MTH1 in platelets, I request to perform immune electron microscopy with anti-MTH1 antibody using isolated platelets. In Figure 1D, please provide appropriate references for tail bleeding assay, and how to define the bleeding time. Figure 1E, please define how to measure the occlusion time, and more detailed information for the images shown in right panels, with definition of dotted lines in the legend, and appropriate scale bars. It is better to show blood vessel with a specific marker. In Figure F, please provide detailed methods how to isolate the thrombus and what are the images in the right panels, in the legend and material and methods, with appropriate scale bars. Please specify the statistical analysis for each comparison in all figure legends, accordingly.

8 . For the figures using IF and confocal imaging, it is recommended to include a panel of morphology of cells or platelets, such as DIC or phase contrast, to provide a better understanding of the localization of MTH1 and well as its abundance to the platelets.

9 . In Figure 5, it would be helpful to include the main phosphoproteins highlighted in the volcano plots to provide a more comprehensive analysis of the phosphoproteome changes upon MTH1 deficiency.

10 . In Figures 3 and 7, it is difficult to recognize the localization of 8-oxo-dG in side or outside of mitochondria. I suggest showing VDAC image alone too, and orthogonal views of merged images to clarify the localization in much more magnified view of high resolution images. Additionally, the authors did not perform RNAase treatments before 8-oxo-dG staining, which could potentially lead to false-positive signals from RNA which is more highly susceptible to oxidation. It would be useful to provide such experiments to rule out this possibility.

11 . The labels of figures should be modified to have a bigger font and follow a uniform style for consistency and clarity.

12 . All bar plots should include individual data points to provide a better understanding of the variability and statistical significance of the results.

13 . Finally, it is recommended to modify the figure titles to highlight the main findings of each figure, and also spell out the abbreviations in each legend, which would help readers to better understand the results.

Reviewer #2 (Remarks to the Author):

The manuscript provides novel data that Nudix hydrolase 1 (NUDT1) in mouse platelets plays a functional role in the hydrolysis of oxidized nucleotides and that the targeted knockout of this gene leads to alterations in platelet functions, for instance under conditions of oxidative stress. In general, the paper is well-designed with multiple interesting findings and at first reading provides a strong case. The experimental approach is strong. However, upon closer inspection the paper also has several major fundamental and experimental shortcomings.

1. Recent and general background information on the gene and enzyme is missing in the manuscript. The current name of the gene is NUDT1 (Nudix hydrolase 1), of which according to current databases the transcripts are indeed amply present in human and mouse megakaryocytes and platelets. Databases confirm the enzyme's function such as reported in the manuscript, but also state that the encoded protein is mainly localized in the cytoplasm, with some in the mitochondria (and little in the nucleus), suggesting that it is involved in the sanitization of mostly cytosolic-derived nucleotide pools for nuclear and to a certain extent for mitochondrial genomes. NUDT1 is a member of the Nudix hydrolase superfamily of about 25 NUDT forms, all containing the NUDIX motif, of which >14 are expressed at reasonable levels in human and mouse platelets. Several members of this superfamily catalyze the hydrolysis of nucleoside diphosphates, including substrates like 8-oxo-dGTP that are a result of oxidative damage, and can induce base mispairing during DNA replication, causing transversions. Disease databases (OMIM) further confirm a role of NUDT1 as anticancer target related to cytotoxicity and DNA damage. Both loss- and gain-of-function mutations have been described in cancer cells (AA 27-156, see UniProt). The UniProt database displays a warning: "A later study indicates that NUDT1 plays a redundant role in eliminating oxidized nucleotides and that it is not essential for cancer cell proliferation and survival (PubMed:28679043)." This remark is consistent with the abundant expression of other NUDT forms in platelets and other tissues, but also raises doubt on a specific localization in platelets in the mitochondria, such as stated in the manuscript. The localization issue needs to be better resolved, although it seems that a mitochondrial localization is not required for explaining the mouse platelet phenotypes. Is there preferred relocation on the transcript or protein level into mitochondria?

2. Related to point 1, it should be discussed to which extent a mitochondrial localization is needed for explaining the platelet phenotypes?

3. Another major concern of the manuscript is the stated 'systematic' difference between thrombin (PAR3/4) and CRP (GPVI) induced platelet responses. For either agonist and for any phenotype dose-response curves are missing. In the various assays a different dose of thrombin or CRP is used, apparently with the latter agonist induced only 40% P-selectin exposure (Figure S4). In addition, different platelet counts are used for the experiments. The paper reads like that the agonist concentrations selected (non-intentional) for each assay needed to support the idea of a difference between GPVI and GPCR signalling. Key missing information are for WT and KO: full thrombin/CRP dose-responses of platelet aggregation, secretion, ROS production, 8-

OHdG accumulation in platelets and mitochondria, etc (Figure 2, 3, 7, Figure S6). Furthermore, finding different effects upon GPVI or thrombin stimulation is insufficient to conclude and extrapolate to GPCR effects: also signalling to ADP, PAR4 and thromboxane receptors should be examined.

4. The paper will be strengthened if the authors apply *in vivo* or *ex vivo* thrombosis experiments under conditions of increased oxygen stress. This will in particular help to understand if the role of the NUDT1 enzyme on platelets is to be seen on the level of replication, transcription and translation (i.e. in megakaryocytes) or on post-translational processes such as ROS-dependent signal transduction. Potentially or even likely, there is a dual role (pre- and post-translational), but also this need to make clear.

5. The suggested apoptotic role of murine NUDT1 needs better explanation, given the abundant evidence for non-apoptotic agonist-induced phosphatidylserine exposure in platelets. In particular the Ca²⁺ signalling points to non-apoptotic, mitochondrial-linked cell death, rather than to apoptosis. Caspase or calpain inhibitor experiments can be performed to discriminate.

6. A different to explain result is the profound prolongation in KO mice of tail bleeding times, versus the only small and partial reduction in the applied models of arterial thrombus formation. This is unusual. Further explanation on this particular finding is needed, as the authors do not report on other blood cell counts, VWF levels and coagulation activity. Has all of this being checked?

7. Figure S5 does not contain quantification, i.e. to compare the amounts of oxidized nucleotides in cytoplasm and mitochondria. Figure S8 is misleading, as given the small amount of mitochondrial DNA, relative oxidation is much easier to occur than in nuclear DNA. What is the oxidation in response to doses of thrombin or CRP?

8. The thrombin-induced phosphoproteome data is interesting, but the authors should be careful in linking the phosphorylation changes of signalling proteins to the changes in mitochondrial proteins. These could be parallel rather than functionally linked events.

9. Some of the data of Figure 6 need further exploration. Why does 3 minutes stimulation with thrombin lead to an almost complete reduction in expression of MT-CO1? Is this this proteasomal, proteolytic. What is high CRP doing?

10. The abstract and discussion need to make clearer that not all platelet phenotypes may be mitochondrial related. Also altered cytoplasmic oxidative stress can contribute to the KO phenotype. Effects of higher CRP doses need to be incorporated.

11. Minor: the native English-speaking co-authors can further help to correct the multiple errors in English language.

Reviewer #3 (Remarks to the Author):

This is a review of a manuscript titled “MTH1 protects platelet mitochondria from oxidative damage and is essential for GPCR-dependent platelet function and thrombosis” authored by Ding et al. In this manuscript, the authors characterized the function of MTH1 in platelets. They use a broad range of different assays and methods to characterize the effects of MTH1 deficiency at a cellular level. Specifically, the authors show that MTH1 deficiency reduced several key platelet functions including mitochondrial metabolism and DNA through ROS in in thrombin-stimulated platelets.

The manuscript is well written, and the data is presented clearly. However,

Major comments: The metabolomics section cannot be reviewed due to lack of information provided in the methods section.

1) The reporting of the metabolomics data is not appropriate for any type of publication. The description of the LC-MS analysis and instrument settings is incomplete. For any journal, but especially for high impact journals, this information is critical for reproducibility. Some important parameters to describe are: volume used for extraction, number of cells, chromatographic gradient, solvents used, column used and temperature, flow rate, voltages, gas flows, scan rate, etc.

2) Additionally, the level of identification for the metabolites is not reported. Were they matched against MS/MS libraries and confirmed with standard reference materials? Or are the identifications based only on accurate mass at MS1 level? There are some metabolites in the supplementary tables with retention times below 1 minute (no retention) that are usually present at very low levels and specialized enrichment and targeted methods are needed for their detection (for example AA derived singling molecules). What is the confidence of those identifications and measurements?

3) How was the metabolomics data processed? What tools and what settings were selected? Has the data been deisotoped and adducts been removed? Based on the volcano plots (Figure 5A) it seems this hasn't been done due to the large number of metabolic features plotted.

Minor comments:

- 1) The volcano plots (figure 5A) are of very low quality and don't show anything relevant for the paper in their current form as no metabolite is labeled.
- 2) Describe what some acronyms stand for (e.g. 8-oxo-dGTP).
- 3) Are there increased levels of lyso-phospholipids that would confirm AA mobilization by cPLA2?

REVIEWER COMMENTS

Reviewer #1 (Remarks to the Author):

Ding et al aimed to investigate the role of the nucleotide pool sanitization enzyme, human MutT Homolog 1 (MTH1), in platelet function and thrombosis. The study found that MTH1 is present in both human and mouse platelets, and deficiency of MTH1 in platelets/megakaryocytes prolonged tail-bleeding time and impaired both arterial and venous thrombus formation. The study also found that MTH1 deficiency significantly reduced platelet aggregation, phosphatidylserine exposure, and calcium mobilization induced by thrombin but not by collagen-related peptide (CRP). Further, MTH1 deficiency impaired platelet metabolism after thrombin stimulation and caused oxidative damage to platelet mitochondrial DNA, reducing the expression of cytochrome c oxidase 1 (MT-CO1) in thrombin-stimulated platelets. Overall, the study provides novel insights into the regulatory role of MTH1 in platelet function and thrombus formation, and supports the potential of MTH1 as a therapeutic target for prevention of thrombotic or cardiovascular diseases.

However, there are several issues that need to be addressed to improve the manuscript:

Major comments

1. English writing is a little bit poor, especially many scientific terminology is wrong, so need to be proofread by English native scientists.

AU RESPONSE: The English writing has been improved and proofread by scientists with English as their native language.

2. Terminology: 8-OH-dG can be used only for name of Kit in Material and Methods. In main text, it should be 8-oxo-7'8'-dyhydro-2'-deoxyguanosine (8-oxo-2'-deoxyguanosine or 8-oxo-dG). Please define 8-oxo-dGTP, 2-OH-dATP, and 8-oxo-dATP, in the Introduction with their full-names and abbreviations, with appropriate references.

AU RESPONSE: 8-OH-dG has been revised to 8-oxo-7'8'-dyhydro-2'-deoxyguanosine (8-oxo-dG). 8-oxo-dGTP, 2-OH-dATP, and 8-oxo-dATP have been defined and appropriate references have been provided.

3. Terminology: page 10, line 17-18: "such as 8-OHdG DNA glycosylase (OGG1)" is wrong, it should be "8-oxoG DNA glycosylase (OGG1)" which excises the oxidized base 8-oxoG, not 8-oxo-dG.

AU RESPONSE: This has been corrected as suggested.

4. Introduction, page 3, line 4; "a double plasma membrane" may not be appropriate

for membranes of mitochondria and nucleus, “two lipid bilayer membranes” may be better.

AU RESPONSE: We have revised this sentence as suggested.

5. Introduction, page 3, line 14-15: “Under normal physiological conditions, the cellular DNA and nucleotide pool are continuously exposed to ROS-induced oxidative damage (11, 12)” may be “Under normal physiological conditions, the cellular DNA and nucleotide pool are continuously exposed to ROS, thereby oxidatively damaged (11, 12)”.

AU RESPONSE: We have revised this sentence as suggested.

6. Introduction, Page 3, line 20: “repairing enzymes such as Human MutT homolog 1(MTH1)” is wrong, because MTH1 is not a repairing enzyme, it is “a nucleotide pool sanitizing enzyme”.

AU RESPONSE: We have revised this sentence as suggested.

7. In Figure 1, it would be important to provide platelet-specific markers to demonstrate the purity of platelets for MTH1 quantification. This would help to exclude the possibility of contamination by other cell types that could affect the interpretation of the results. To demonstrate mitochondrial localization of MTH1 in platelets, I request to perform immune electron microscopy with anti-MTH1 antibody using isolated platelets. In Figure 1D, please provide appropriate references for tail bleeding assay, and how to define the bleeding time. Figure 1E, please define how to measure the occlusion time, and more detailed information for the images shown right panels, with definition of dotted lines in the legend, and appropriate scale bars. It is better to show blood vessel with a specific marker. In Figure F, please provide detailed methods how to isolate the thrombus and what are the images in the right panels, in the legend and material and methods, with appropriate scale bars. Please specify the statistical analysis for each comparison in all figure legends, accordingly.

AU RESPONSE: We thank the reviewer for this valuable comment. The purity of platelets for MTH1 quantification has been evaluated using platelet-specific marker CD41. The extent of contamination by leukocytes has also been assessed by probing for leukocyte-specific marker CD45, which was shown to be absent from the platelet preparations (Figure I).

Figure I. Expression of CD41 and CD45. Mononuclear cells and platelets were

isolated from human or mouse blood to measure the expression of CD41 and CD45 (representative of three independent experiments).

To clarify this, the new data have been added to Figure 1a with incorporation of the following information into the text on page 4 as follows:

“To exclude the potential contamination of leukocytes, we measured the expression of CD45 (a leukocyte maker) in our preparations of isolated platelets and could not detect CD45 expression (Figure 1a), indicating that leukocytes were below detectable levels in the prepared platelets.”

As suggested by the reviewer, immune electron microscopy was performed, and the results (**Figure II**) showed that MTH1 was localized to the mitochondria of platelets.

Figure II. Localization of MTH1 in platelets. Platelets were isolated from wild-type mouse and then fixed followed by labelling with MTH1 antibody and then with the secondary antibody. M indicates mitochondria and arrows indicates the positive expression of MTH1. Left panel: x 12000 and right panel: x 30000.

This new data has been added to the Figure 1c with incorporation of the following information into the text on page 5 as follows:

“Consistently, the immune electron microscopy analysis also showed the localization of MTH1 protein to the mitochondria of platelets (Figure 1d).”

The references for the tail bleeding assay have been provided and the bleeding time was defined as the time at which bleeding had visibly ceased. This information has been included in the text.

The occlusion time (time to occlusion) was defined as the time from FeCl₃ application to the cessation of blood flow in the vessel as defined by the flow probe. The dotted lines have been defined in the legend and the scale bars have been provided. In the FeCl₃-induced arterial thrombosis model, the mesenteric arterioles were surgically exposed and visualized using microscopy and treated with FeCl₃ to initiate the arterial thrombus formation, the arterial vessel walls (dotted lines in the images) can be visualized directly under a microscope, therefore, the blood vessels might not be required to be labelled.

Detailed method on how to isolate the venous thrombus was provided in the method section as follows:

“After 24 hours of ligation, mice were humanely sacrificed and the abdominal cavity was surgically opened. The IVC and associated thrombi were removed and the thrombus length and weight were measured.”

The images in the right panel have been explained in the legend and the scale bar has also been provided.

The statistical analysis for each comparison has been provided in all figure legends.

8. For the figures using IF and confocal imaging, it is recommended to include a panel of morphology of cells or platelets, such as DIC or phase contrast, to provide a better understanding of the localization of MTH1 and well as its abundance to the platelets.

AU RESPONSE: We thank the reviewer for this comment and agree with the reviewer that it would be more informative to include a panel of the morphology of platelets. However, the fluorescence-labelled antibody and the fluorescence quencher used in the IF and confocal imaging experiments interfere with the observation and analysis of platelet morphology under the differential interference contrast (DIC) microscopy. Therefore, it is not possible to assess the morphology of platelets in the IF and confocal imaging experiments.

9. In Figure 5, it would be helpful to include the main phosphoproteins highlighted in the volcano plots to provide a more comprehensive analysis of the phosphoproteome changes upon MTH1 deficiency.

AU RESPONSE: We thank the reviewer for this comment. The detailed information on the changes of phosphoproteins has been provided in the Supplemental file S5. The purpose of the volcano plots is just to provide an overview of the profiles of phosphoproteins changes between control and MTH1-deficient platelets after stimulation.

10. In Figures 3 and 7, it is difficult to recognize the localization of 8-oxo-dG in side or outside of mitochondria. I suggest showing VDAC image alone too, and orthogonal views of merged images to clarify the localization in much more magnified view of high resolution images. Additionally, the authors did not perform RNAase treatments before 8-oxo-dG staining, which could potentially lead to false-positive signals from RNA which is more highly susceptible to oxidation. It would be useful to provide such experiments to rule out this possibility.

AU RESPONSE: We thank the reviewer for this comment and have included a control VDAC image and showed the localization of 8-oxo-dG inside mitochondria. Meanwhile, the magnified view of the merged images was shown in Figures 3A and 7C and the original images were shown in the supplemental Figure S7 and S15.

To exclude the false-positive signals from RNA, we performed RNase treatment before 8-oxo-dG staining as suggested by the reviewer. The results showed no difference of the positive 8-oxo-dG signal in platelets treated with RNase compared to that in platelets treated with vehicle (**Figure III** and **Figure IV**), indicating that the positive 8-oxo-dG signal does not include the contamination of RNA oxidation.

Figure III. 8-oxo-dG staining in mouse platelets. Platelets isolated from MTH1^{-/-} mice and stimulated with thrombin (1 U/ml) for 3 min followed by treatment with RNase A (0.02 mg/ml) for 30 minutes at 37°C. After that, platelets were stained with antibodies against with 8-oxo-dG or VDAC and the subsequent with fluorescence-labelled secondary antibodies. Then, the staining was visualized under a confocal microscopy. Representative results from three independent experiments were shown (mean ± SE, n=3).

Figure IV. 8-oxo-dG staining in human platelets. Human platelets were pre-treated with 5 μM TH588 for 1 h at 37 °C and then stimulated with thrombin (1 U/ml) for 3 min followed by treatment with RNase A (0.02 mg/ml) for 30 minutes at 37°C. After that, platelets were stained with antibodies against with 8-oxo-dG or VDAC and then with secondary antibodies followed by observation of the staining under a confocal microscopy. Representative results from four independent experiments were shown (mean ± SE, n=4).

To clarify this, the new data have been added to Figure S8 or Figure S16 with incorporation of the following information into the text on page 7 or page 14 as follows:

“To exclude any potential false-positive signals from RNA, we performed RNase treatment before 8-oxo-dG staining and found no differences in the

positive 8-oxo-dG signal in platelets treated with RNase compared to platelets treated with vehicle (Figure S8), indicating that contamination of RNA oxidation does not contribute to the positive 8-oxo-dG signal.”

“Further, the positive 8-oxo-dG signal in human platelets did not include any RNA oxidation signal contamination as shown by no difference between positive 8-oxo-dG signal in human platelets treated with RNase versus that in platelets treated with vehicle (Figure S16).”

11. The labels of figures should be modified to have a bigger font and follow a uniform style for consistency and clarity.

AU RESPONSE: The labels of figures have been modified and a uniform style has been adopted.

12. All bar plots should include individual data points to provide a better understanding of the variability and statistical significance of the results.

AU RESPONSE: Individual data points have been provided for all bar plots.

13. Finally, it is recommended to modify the figure titles to highlight the main findings of each figure, and also spell out the abbreviations in each legend, which would help readers to better understand the results.

AU RESPONSE: The figure titles have been modified to highlight the main findings. The abbreviations in the legend have been defined.

Reviewer #2 (Remarks to the Author):

The manuscript provides novel data that Nudix hydrolase 1 (NUDT1) in mouse platelets plays a functional role in the hydrolysis of oxidized nucleotides and that the targeted knockout of this gene leads to alterations in platelet functions, for instance under conditions of oxidative stress. In general, the paper is well-designed with multiple interesting findings and at first reading provides a strong case. The experimental approach is strong. However, upon closer inspection the paper also has several major fundamental and experimental shortcomings.

1. Recent and general background information on the gene and enzyme is missing in the manuscript. The current name of the gene is NUDT1 (Nudix hydrolase 1), of which according to current databases the transcripts are indeed amply present in human and mouse megakaryocytes and platelets. Databases confirm the enzyme's function such as reported in the manuscript, but also state that the encoded protein is mainly localized in the cytoplasm, with some in the mitochondria (and little in the nucleus), suggesting that it is involved in the sanitization of mostly cytosolic-derived nucleotide pools for nuclear and to a certain extent for mitochondrial genomes. NUDT1 is of member of the Nudix hydrolase superfamily of about 25 NUDT forms, all containing the NUDIX motif, of which >14 are expressed at reasonable levels in human and mouse platelets. Several members of this superfamily catalyse the hydrolysis of nucleoside diphosphates, including substrates like 8-oxo-dGTP that are a result of oxidative damage, and can induce base mispairing during DNA replication, causing transversions. Disease databases (OMIM) further confirm a role of NUDT1 as anticancer target related to cytotoxicity and DNA damage. Both loss- and gain-of-function mutations have been described in cancer cells (AA 27-156, see UniProt). The UniProt database displays as a warning: "A later study indicates that NUDT1 plays a redundant role in eliminating oxidized nucleotides and that it is not essential for cancer cell proliferation and survival (PubMed:28679043)." This remark is consistent with the abundant expression of other NUDT forms in platelets and other tissues, but also raises doubt on a specific localization in platelets in the mitochondria, such as stated in the manuscript. The localization issue needs to be better resolved, although it seems that a mitochondrial localization is not required for explaining the mouse platelet phenotypes. Is there preferred relocation on the transcript or protein level into mitochondria?

AU RESPONSE: We thank the reviewer for this comment. The most recent and general background information on the gene and enzyme has been provided in the Introduction section. In addition, the localization of MTH1 in platelets in the mitochondria has been further verified and confirmed by immune-electron microscopy analysis (**Figure II**).

Figure II. Localization of MTH1 in platelets. Platelets were isolated from wild-type mouse and then fixed followed by labelling with MTH1 antibody and then with the secondary antibody. M indicates mitochondria and arrows indicates the positive expression of MTH1. Left panel: x 12000 and right panel: x 30000.

2. Related to point 1, it should be discussed to which extent a mitochondrial localization is needed for explaining the platelet phenotypes?

AU RESPONSE: We thank the reviewer for this comment. The fractional immunoblotting approach and immune-electron microscopy analysis confirmed the localization of MTH1 in the mitochondria of platelets, which helps explain the observed phenotypes of MTH1-deficient platelets, including impaired mitochondrial ATP production (Figure 2h), increased oxidative damage to mitochondrial DNA (Figure 3a), as well as the reduced expression of proteins encoded by mitochondrial DNA (Figure 6c) in response to thrombin. The distinct effect of thrombin versus CRP on the function of MTH1-deficient platelets would further support the role of mitochondrial MTH1 in GPCR-dependent platelet signaling as shown by thrombin-mediated generation of mitochondrial ROS, which can cause oxidative damage to platelet mitochondria and other platelet phenotypes. However, as we mentioned in the following comments, it is also possible that the altered cytoplasmic oxidative stress indicated by impaired ROS-sensitive signaling pathways (p38 kinase and AKT) might also contribute to the observed phenotype of MTH1-deficient platelets.

3. Another major concern of the manuscript is the stated ‘systematic’ difference between thrombin (PAR3/4) and CRP (GPVI) induced platelet responses. For either agonist and for any phenotype dose-response curves are missing. In the various assays a different dose of thrombin or CRP is used, apparently with the latter agonist induced only 40% P-selectin exposure (Figure S4). In addition, different platelet counts are used for the experiments. The paper reads like that the agonist concentrations selected (non-intentional) for each assay needed to support the idea of a difference between GPVI and GPCR signalling. to mark the inter-agonist differences. Key missing information are for WT and KO: full thrombin/CRP dose-responses of platelet aggregation, secretion, ROS production, 8-OHdG accumulation in platelets and mitochondria, etc (Figure 2, 3,

7, Figure S6). Furthermore, finding different effects upon GPVI or thrombin stimulation is insufficient to conclude and extrapolate to GPCR effects: also signalling to ADP, PAR4 and thromboxane receptors should be examined.

AU RESPONSE: We thank the reviewer for this comment. In the various assays assessing platelet function, different doses of thrombin and CRP were generally used to accommodate different detection sensitivities between assays. For example, the doses of agonists used for platelet aggregation and granule release are usually lower than those for western blot assays or immunofluorescent staining assays. Further, several assays call for different numbers of platelets. In this study, we carefully unified the respective platelet counts within an assay in order to compare data from control platelets and MTH1-deficient platelets before and after stimulation. All the comparisons between control platelets *versus* MTH1-deficient platelets in our study were made before and after the same agonist stimulation, rather than the comparison of CRP *versus* thrombin since these agonists are known to differentially activate platelet signaling pathways.

As suggested by the reviewer, the dose-response curves of platelet agonists have been provided including the full thrombin/CRP dose-responses for platelet aggregation and secretion. As seen in **Figure V**, stimulation with higher doses of thrombin (0.05 U/ml) or U46619 (1 μ M) eliminated platelet aggregation and ATP release differences between control and MTH1-deficient platelets. Further, higher doses of CRP (1 μ g/ml) or thrombin (0.05 U/ml) achieved the full platelet aggregation and secretion responses as demonstrated by more than 85% of platelet aggregation and ATP release (dense granule release) in control platelets without a difference between control and MTH1-deficient platelets. Our data showed no difference of platelet aggregation and secretion in response to either a low or high doses of CRP between control and MTH1-deficient platelets, confirming that MTH1 deficiency does not affect GPVI-dependent platelet aggregation and ATP secretion.

To clarify this, the new data have been added to Figure S5 with incorporation of the following information into the text on page 6 as follows:

“We then evaluated MTH1’s effect on platelet function in vitro and showed that MTH1 deficiency did not affect CRP-mediated platelet aggregation or ATP release at low (0.1 μ g/ml) (Figure 2a) or high dose (1 μ g/ml) (Figure S5). However, MTH1 deficiency resulted in significant inhibition of platelet aggregation and ATP production in response to the low dose of thrombin (0.01 U/ml) (Figure 2b) or U46619 (0.3 μ M) (Figure 2c), but not to a higher dose of thrombin (0.05 U/ml) (Figure S4) or U46619 (1 μ M) (Figure S5), suggesting that MTH1 might regulate GPCR-dependent platelet function.”

Figure V. Platelet aggregation and ATP release. Washed platelets from MTH1^{fl/fl} or MTH1^{-/-} mice were stimulated with CRP (a), thrombin (b) or U46619 (c) to measure platelet aggregation and ATP release. Data were shown as mean ± SE (n = 3 independent experiments).

For human platelet studies, the dose-response curves of platelet agonists are provided within inclusion of the full thrombin/CRP dose-responses of platelet aggregation and secretion. As seen in **Figure VI**, no difference of platelet aggregation and ATP secretion was observed between vehicle-treated and TH-588 treated platelets in response to high dose of thrombin (0.1 U/ml) and CRP (2 µg/ml).

To clarify this, the new data have been added to Figure S14 with incorporation of the following information into the text on page 14 as follows:

“Our results showed that treatment with TH-588 to inhibit MTH1 significantly reduced platelet aggregation and ATP production in response to a low (0.04 U/ml) (Figure 7a) but not to a high dose (0.1 U/ml) (Figure S13) of thrombin. However, TH588 did not alter platelet aggregation and ATP secretion in response to CRP (0.5 or 2 µg/ml) (Figure 7b, Figure S14 respectively).”

Figure VI. Platelet aggregation and ATP release in human platelets. Washed human platelets were pre-treated with 5 µM TH588 (MTH1 inhibitor) for 1 h at 37 °C followed by measuring platelet aggregation and ATP release in response to thrombin (0.1 U/ml) (mean ± SD, n = 3, unpaired student's t-test) (a) or CRP (2 µg/ml) (mean ± SD, n = 3, unpaired student's t-test).

To provide more evidence of the effect of MTH1 on GPCR signaling, we also investigated the signaling from ADP and thromboxane receptors as suggested by the reviewer. As seen in **Figure VII**, MTH1 deficiency significantly increased the phosphorylation of PLCβ3 at Ser1105 and decreased the phosphorylation of AKT at Ser473 in platelets stimulated with GPCR agonist, ADP or U46619, further supporting a role for MTH1 in GPCR signal transduction.

Figure VII. The phosphorylation level of PLCβ3 and AKT. MTH1^{fl/fl} or MTH1^{-/-} platelets were treated with ADP (10 µM) or U46619 (1 µM) for the indicated time

points and then the phosphorylation level of PLC β 3 (Ser1105) and AKT (Ser473) was measured. The data were quantified based on three independent experiments (mean \pm SD, n = 3, two-way ANOVA with Sidak multiple comparisons test).

To clarify this, the new data have been added to Figure S10 with incorporation of the following information into the text on page 10 as follows:

“To evaluate whether MTH1 deficiency affects other GPCR signaling pathways, we evaluated phosphorylation of PLC β 3 and AKT in platelets treated with ADP or U46619. The results showed that MTH1 deficient platelets had significantly increased levels of phosphorylation of PLC β 3 at Ser1105 and decreased phosphorylation of AKT at Ser473 after stimulation with GPCR agonists, ADP or U46619 (Figure S10), further supporting a role for MTH1 in GPCR signaling transduction.”

4. The paper will be strengthened if the authors apply in vivo or ex vivo thrombosis experiments under conditions of increased oxygen stress. This will in particular help to understand if the role of the NUDT1 enzyme on platelets is to be seen on the level of replication, transcription and translation (i.e. in megakaryocytes) or on post-translational processes such as ROS-dependent signal transduction. Potentially or even likely, there is a dual role (pre- and post-translational), but also this need to make clear.

AU RESPONSE: We thank the reviewer for this comment. Regarding the role of this enzyme on platelets at the level of replication, transcription, translation or post-translation processes, our data from megakaryocytes and platelets have demonstrated that oxidative stress induces much more mitochondrial DNA damage in the case of NUDT1 enzyme deficiency (new Figure S12), which may affect the replication, transcription and translation of megakaryocyte mitochondrial genes, causing possible alterations (such as inclusion of differentially expressed mitochondrial gene-encoded proteins) to the platelets that are produced. In platelet studies, our results showed that stimulation of mitochondrial ROS generation by thrombin results in significantly reduced gene and protein expression of mitochondrial gene-encoded MT-CO1 (Figure 6c and 6d) in MTH1-deficient platelets. One possible reason might be related to the higher guanine content in MT-CO1 DNA relative to other mtDNAs (Figure 6e), making this gene more susceptible to oxidative damage during DNA replication in activated MTH1-deficient platelets. This may lead to impaired translation and protein synthesis, consistent with our quantitative phosphoproteomic analysis that shows a decreased phosphorylation of Slirp, which regulates mitochondrial protein synthesis. Although platelets are anucleate, they harbor a diverse set of RNAs and possess multiple post-transcriptional and protein synthesis regulatory pathways, which are critical for rapid responses to changing environmental properties (Neu et al., *Int J Mol Sci.* 2020 and Zimmerman et al., *Arterioscler Thromb Vasc Biol.* 2008). Regarding the post-translational processes, our data also showed impaired ROS-dependent signaling pathways such as phosphorylation of p38 and AKT in knockout platelets after stimulation (Figure 5d). Therefore, from the data we obtained, this enzyme plays a dual role in platelet function.

To clarify this, the following information has been added into the text on page 16:

“Considering the significantly impaired ROS-dependent signaling pathways (reduced phosphorylation of p38 and AKT) in MTH1-deficient platelets after stimulation, it seems that MTH1 plays a dual role (pre- and post-translation) in platelet function.”

5. The suggested apoptotic role of murine NUDT1 needs better explanation, given the abundant evidence for non-apoptotic agonist-induced phosphatidylserine exposure in platelets. In particular the Ca²⁺ signalling points to non-apoptotic, mitochondrial-linked cell death, rather than to apoptosis. Caspase or calpain inhibitor experiments can be performed to discriminate.

AU RESPONSE: We thank the reviewer for this comment and agree that agonist-induced PS exposure events in platelets are distinct from apoptosis-induced PS exposure. Specifically in our study, the reduced Ca²⁺ signaling induced by thrombin indicates the triggering of a non-apoptotic mitochondrial-linked cell death pathway. Since thrombin in isolation is a relatively weak proapoptotic platelet agonist (Leytin et al., *Br J Haematol.* 2007 and Gyulkhandanyan et al., *Br J Haematol.* 2013), to evaluate the role of NUDT1 in platelet apoptosis, non-physiological agents such as the BH3 mimetic ABT-737 could be utilized as a comparator. However, all these reagents have additional off-target effects that may interfere with interpretation of data. As the main focus of our study was to assess GPCR-dependent platelet function using thrombin as a GPCR agonist, investigation of platelet apoptosis using other agents, is beyond the scope of our current manuscript. To avoid any confusion, we decided to remove the caspase-3 data and plan to perform further experiments to investigate the role of NUDT1 in platelet apoptosis in the future.

6. A different to explain result is the profound prolongation in KO mice of tail bleeding times, versus the only small and partial reduction in the applied models of arterial thrombus formation. This is unusual. Further explanation on this particular finding is needed, as the authors do not report on other blood cell counts, VWF levels and coagulation activity. Has all of this being checked?

AU RESPONSE: We thank the reviewer for this comment. Regarding hemostasis and thrombosis, these two processes are distinct involving variable contributions to signaling pathways, that then differentially modulate integrin α IIB β 3 bidirectional signaling (Estevez et al., *Arterioscler Thromb Vasc Biol.* 2015; Huang et al., *J Hematol Oncol.* 2019.). Selectively targeting α IIB β 3 outside-in signaling has been postulated as a therapeutic avenue to inhibit thrombosis while maintaining hemostasis in animal models (Shen et al., *Nature.* 2013), in line with the critical role of α IIB β 3 outside-in signaling in thrombus formation. In our study, MTH1 deficiency did not affect platelet spreading (Figure S18) or clot retraction (Figure S19), two processes regulated by α IIB β 3 outside-in signaling, implying that MTH1 may not play a role in α IIB β 3 outside-in signaling, thus explaining the small or partial reduction of arterial thrombus

formation in mice.

To clarify this, the following information has been added into the text on page 14:

“Interestingly, MTH1 deficiency caused a profound prolongation of tail bleeding time, in contrast to the relatively small/partial reduction in the arterial thrombus formation observed. Hemostasis and thrombosis events are distinct, involving variable contributions to signaling pathways, that then differentially modulate integrin α IIB β 3 bidirectional signaling (Estevez et al., *Arterioscler Thromb Vasc Biol.* 2015; Huang et al., *J Hematol Oncol.* 2019). Selectively targeting α IIB β 3 outside-in signaling has been postulated as a therapeutic avenue to inhibit thrombosis while maintaining hemostasis in animal models (Shen et al., *Nature.* 2013), in line with the critical role of α IIB β 3 outside-in signaling in thrombus formation. In our study, MTH1 deficiency did not affect platelet spreading (Figure S18) or clot retraction (Figure S19), two processes regulated by α IIB β 3 outside-in signaling, implying that MTH1 may not play a role in α IIB β 3 outside-in signaling, thus explaining the small or partial reduction of arterial thrombus formation in mice.”

As suggested by the reviewer, we have measured the number of white blood cells and red blood cells as well as VWF levels and coagulation activity (the activated partial thromboplastin time). The results showed no significant differences of these parameters between control mice and MTH1-deficient mice (**Figure VIII**).

Figure VIII. The relative numbers of white and red blood cells, VWF levels and coagulation activity. Blood was isolated from MTH1^{fl/fl} or MTH1^{-/-} mice and white (a) and red (b) blood cells were quantified. In addition, plasma VWF levels (c) and the activated partial thromboplastin time (d) were ascertained.

To clarify this, the following information has been added into the text on page 5:

“To evaluate the influence of other blood cells and coagulation activity on the hemostasis and thrombosis, we measured the number of white blood

cells and red blood cells as well as VWF levels and coagulation activity (the activated partial thromboplastin time) and the results showed that no significant differences in any of these parameters between control mice and MTH1-deficient mice (Figure S4).”

7. Figure S5 does not contain quantification, i.e. to compare the amounts of oxidized nucleotides in cytoplasm and mitochondria. Figure S8 is misleading, as given the small amount of mitochondrial DNA, relative oxidation is much easier to occur than in nuclear DNA. What is the oxidation in response to doses of thrombin or CRP?

AU RESPONSE: We thank the reviewer for this comment.

In Figure S5 (new Figure S7), Panel A shows the isotype control staining for the 8-oxodG antibody and Panel B shows the original images of 8-oxodG antibody staining from Figure 3A (Figure 3A shows the enlarged images of Panel B in the Figure S5). The quantification was already shown in Figure 3A.

We agree with the reviewer that mitochondrial DNA is much susceptible to oxidation than nuclear DNA. However, the purpose of Figure S8 was to evaluate the effect of MTH1 on the oxidation damage of mitochondrial DNA *versus* nuclear DNA. Our results showed that under the conditions of oxidative stress, wild-type or control megakaryocytes mitochondrial DNA was more susceptible to oxidative damage as compared with nuclear DNA, which is consistent with a previous study (Yakes FM et al., *P Natl Acad Sci USA* 1997). However, when comparing mitochondrial DNA damage between genotypes, there was significantly more oxidative damage to mitochondrial DNA from MTH1-deficient megakaryocytes relative to control megakaryocytes, indicating that mitochondrial DNA oxidative damage was enhanced in the setting of MTH1 deficiency.

To clarify this, the following information has been added into the text on page 13:

“However, when comparing mitochondrial DNA damage between genotypes, there was significantly more oxidative damage to mitochondrial DNA from MTH1-deficient megakaryocytes relative to control megakaryocytes, indicating that mitochondrial DNA oxidative damage was enhanced in the setting of MTH1 deficiency.”

In Figure S8 (new Figure S13), megakaryocytes were cultured and exposed to H₂O₂ to induce oxidative stress to investigate the role of MTH1 in the oxidative damage of mitochondrial DNA or nuclear DNA. Since megakaryocytes were used in this assay rather than platelets, doses of CRP or thrombin was not performed.

8. The thrombin-induced phosphoproteome data is interesting, but the authors should be careful in linking the phosphorylation changes of signalling proteins to the changes in mitochondrial proteins. These could be parallel rather than functionally linked events.

AU RESPONSE: We thank the reviewer for this valuable comment and agree that the phosphorylation changes of signaling proteins should be interpreted with caution and may not be linked directly to changes of mitochondrial proteins. This has been stated

in the revised manuscript in the text on page 16 as follows:

“However, whilst the phosphorylation changes of Slirp and the changes in mitochondrial protein MT-CO1 were co-incidental, further work will be required to demonstrate that these two events are functionally linked events.”

9. Some of the data of Figure 6 need further exploration. Why does 3 minutes stimulation with thrombin lead to an almost complete reduction in expression of MT-CO1? Is this this proteasomal, proteolytic. What is high CRP doing?

AU RESPONSE: We thank the reviewer for this comment. As shown in Figure 3D-F, stimulation with thrombin for 3 minutes triggers the production of mitochondrial ROS, which causes the increase of oxidative damage to mitochondrial DNA in the case of MTH1 deficiency (Figure 3A), leading to impaired replication, transcription and translation of mitochondrial genes, especially MT-CO1 gene with a higher guanine content relative to other mitochondrial genes (Figure 6F), and subsequent reduced gene and protein expression.

The signaling pathways activated by thrombin versus CRP treatment are quite molecularly distinct. Hence the treatment of MTH1-deficient platelets with a high dose of CRP (5 $\mu\text{g/ml}$) did not affect the expression of proteins encoded by either mitochondrial DNA (ND1 and MT-CO1) or nuclear DNA (NDUFV1 and UQCRC2) (**Figure IX**), indicating that CRP did not trigger the oxidative damage to mitochondrion, consistent with the absence of mitochondrial ROS in CRP-stimulated platelets (Figure 3d).

Figure IX. Expression of proteins encoded by mitochondrial DNA or nuclear DNA. Platelets were isolated from MTH1^{fl/fl} or MTH1^{-/-} mice and treated without (-) or with (+) CRP (5 $\mu\text{g/ml}$) for 3 mins followed by measuring the expression of ND1, MT-CO1, NDUFV1, and UQCRC2 (representative of three independent experiments) (mean \pm

SD, n = 3, two-way ANOVA with Sidak multiple comparisons test).

To clarify this, the new data have been added to Figure S12 with incorporation of the following information into the text on page 12 as follows:

“However, no significant changes of the expression of proteins encoded by either mitochondrial DNA (ND1 and MT-CO1) or nuclear DNA (NDUFV1 and UQCRC2) were detected in platelets stimulated with CRP (5 µg/ml) (Figure S12), indicating that GPVI engagement and signaling does not trigger the observed oxidative damage to mitochondrion, which is consistent with the lack of mitochondrial ROS in CRP-stimulated platelets (Figure 3d).”

10. The abstract and discussion need to make clearer that not all platelet phenotypes may be mitochondrial related. Also altered cytoplasmic oxidative stress can contribute to the KO phenotype. Effects of higher CRP doses need to be incorporated.

AU RESPONSE: We thank the reviewer for this comment and agree that altered cytoplasmic oxidative stress might also contribute to the phenotype of MTH1-deficient mice. This observation as well as a comment that not all observed platelet phenotypes may be mitochondrial in basis has been stated in the revised manuscript. In addition, the effects of higher CRP doses were also incorporated.

11. Minor: the native English-speaking co-authors can further help to correct the multiple errors in English language.

AU RESPONSE: The errors in English language have been corrected.

Reviewer #3 (Remarks to the Author):

This is a review of a manuscript titled “MTH1 protects platelet mitochondria from oxidative damage and is essential for GPCR-dependent platelet function and thrombosis” authored by Ding et al. In this manuscript, the authors characterized the function of MTH1 in platelets. They use a broad range of different assays and methods to characterize the effects of MTH1 deficiency at a cellular level. Specifically, the authors show that MTH1 deficiency reduced several key platelet functions including mitochondrial metabolism and DNA through ROS in in thrombin-stimulated platelets.

The manuscript is well written, and the data is presented clearly. However,

Major comments: The metabolomics section cannot be reviewed due to lack of information provided in the methods section.

1. The reporting of the metabolomics data is not appropriate for any type of publication. The description of the LC-MS analysis and instrument settings is incomplete. For any journal, but especially for high impact journals, this information is critical for reproducibility. Some important parameters to describe are: volume used for extraction, number of cells, chromatographic gradient, solvents used, column used and temperature, flow rate, voltages, gas flows, scan rate, etc.

AU RESPONSE: We thank the reviewer for this valuable comment and have inserted detailed information on the metabolomics assay into the text on page 22 as follows:

“After each treatment, suspension of platelets (1×10^7) were washed with PBS and the metabolites were extracted by addition of 800 μ l of cold methanol/acetonitrile (1:1, v/v) to remove the protein. The mixture was centrifuged at 14000 x g for 5min. The isolated supernatant was lyophilized and the residue dissolved in 100 μ l acetonitrile/water (1:1, v/v) solvent for LC-MS analysis performed with an UHPLC (1290 Infinity LC, Agilent Technologies) coupled to a quadrupole time-of-flight (AB Sciex TripleTOF 6600; Shanghai Applied Protein Technology Co., Ltd) mass spectrometer.

The sample was separated on an ACQUITY UPLC BEH HILIC column (2.1 mm x 100 mm, 1.7 μ m) (Waters, Ireland) at 25°C with a flow rate of 0.5 mL/min and injection volume of 2 μ L. The mobile phase composition A: water + 25 mM ammonium acetate + 25 mM ammonia water, B: acetonitrile; the gradient elution program was: 95% B for 0-0.5 min; 95%-65% B for 0.5-7 min; 65%-40% B for 7-8 min; 40% B for 8-9 min; 40%-95% B for 9-9.1 min; 95% B for 9.1-12 min. After the samples were separated by Agilent 1290 Infinity LC UHPLC, mass spectrometry was carried out by Triple TOF 6600 mass spectrometer (AB SCIEX), and the positive ion and negative ion modes of electrospray ionization (ESI) were used for detection. The ESI source setting parameters are set as follows: atomizing gas auxiliary heating gas 1 (Gas1): 60, auxiliary heating gas 2

(Gas2): 60, curtain gas (CUR): 30psi, ion source temperature: 600°C, IonSpray Voltage Floating (ISVF) \pm 5500 V (both positive and negative modes). In MS only acquisition, the instrument was set to acquire over the m/z range of 60-1000 Da with accumulation time of 0.20 s/spectra. In auto MS/MS acquisition, it was set to acquire over the m/z range of 25-1000 Da with accumulation time of 0.05 s/spectra. The MS spectras were acquired in data-dependent acquisition mode (IDA), and peak intensity value screening mode was used with declustering potential (DP) of \pm 60 V (positive and negative two modes) and collision energy of 35 \pm 15 eV. IDA settings were designed as the dynamical exclusion of the isotope ion within the range of 4 Da and collection of 10 fragment spectra for each scan.”

2. Additionally, the level of identification for the metabolites is not reported. Were they matched against MS/MS libraries and confirmed with standard reference materials? Or are the identifications based only on accurate mass at MS1 level? There are some metabolites in the supplementary tables with retention times below 1 minute (no retention) that are usually present at very low levels and specialized enrichment and targeted methods are needed for their detection (for example AA derived singling molecules). What is the confidence of those identifications and measurements?

AU RESPONSE: We thank the reviewer for this comment. The metabolite structure identification was based on accurate mass matching (m/z value < 25 ppm) and secondary spectral matching methods, and annotation levels (0-2) described by the Metabolic Standards Initiative (MSI) were used to search an in-house database (Shanghai Applied Protein Technology Co., Ltd.) established with authentic standards with the scoring threshold is \geq 0.7. This information has been added to the method section in text on page 24 as follows:

“The metabolite structure identification was based on accurate mass matching (m/z value < 25 ppm) and secondary spectral matching methods, and annotation levels (0-2) described by the Metabolic Standards Initiative (MSI) were used to search an in-house database (Shanghai Applied Protein Technology Co., Ltd.) established with authentic standards where the scoring threshold was \geq 0.7.”

We agree with the reviewer that some specialized enrichment and targeted methods are required for the detection the metabolites with retention times less than 1 minutes. However, all of these identified metabolites showed significant differences between two groups based on the OPLS-DA VIP>1 and P value < 0.05. In addition, the main purpose of the non-targeted metabolomics assay was to provide a comprehensive metabolomic profile of MTH1-deficient platelets broadly. Therefore, except for the analysis of the metabolomics data, in vitro experiments are still required to confirm whether there is indeed a significant difference between metabolites of interest. In our study, we found a significantly reduced release of AA (Figure 4d) and thromboxane B2 (an inactive metabolite/product of thromboxane A2) (Figure 4e) from thrombin-stimulated MTH1-deficient platelets compared to control platelets, which was

consistent with the observed decreased arachidonic acid (AA) metabolism in MTH1-deficient platelets after stimulation in the pathway-based analysis of metabolic changes.

3. How was the metabolomics data processed? What tools and what settings were selected? Has the data been deisotoped and adducts been removed? Based on the volcano plots (Figure 5A) it seems this hasn't been done due to the large number of metabolic features plotted.

AU RESPONSE: We thank the reviewer for this comment. The raw MS data (wiff.scan files) were converted into mzXML format using ProteoWizard MSConvert before importing into XCMS software. For peak picking, the settings were: centWave m/z = 25 ppm, peakwidth = c (10, 60), prefilter = c (10, 100). For peak grouping, bw = 5, mzwid = 0.025, minfrac = 0.5 were applied. CAMERA (Collection of Algorithms of MEtabolite pRofile Annotation) was applied for annotation of isotopes and adducts. In the extracted ion features, only the variables having > 50% of the nonzero measurement values in at least one group were kept. The metabolite structure identification was based on accurate mass matching (m/z value < 25 ppm) and secondary spectral matching methods, and annotation levels (0-2) described by the Metabolic Standards Initiative (MSI) was used to search an in-house database (Shanghai Applied Protein Technology Co., Ltd.) established with authentic standards with the scoring threshold is ≥ 0.7 . During the data processing, the isotopes were not analyzed and the adducts were not removed, the corresponding adducted ion forms will be shown in the results.

To clarify this, the following information has been added into the text on page 23:

“The raw MS data (wiff.scan files) were converted into mzXML format using ProteoWizard MSConvert before importing into XCMS software. For peak picking, the settings were: centWave m/z = 25 ppm, peak width = c (10, 60), prefilter = c (10, 100). For peak grouping, bw = 5, mzwid = 0.025, minfrac = 0.5 were applied. CAMERA (Collection of Algorithms of MEtabolite pRofile Annotation) was applied for annotation of isotopes and adducts. In the extracted ion features, only the variables having > 50% of the nonzero measurement values in at least one group were kept. The metabolite structure identification was based on accurate mass matching (m/z value < 25 ppm) and secondary spectral matching methods, and annotation levels (0-2) described by the Metabolic Standards Initiative (MSI) was used to search an in-house database (Shanghai Applied Protein Technology Co., Ltd.) established with authentic standards with the scoring threshold is ≥ 0.7 . During the data processing, the isotopes were not analyzed and the adducts were not removed, the corresponding adducted ion forms will be shown in the results.”

Minor comments:

1) The volcano plots (figure 5A) are of very low quality and don't show anything relevant for the paper in their current form as no metabolite is labeled.

AU RESPONSE: We thank the reviewer for this comment. The purpose of the volcano plots was to provide a broad overview of the breadth of phosphoproteins changes between control and MTH1-deficient platelets after stimulation. A high resolution version of Figure 5A has been provided and the detailed information on relative changes to phosphoproteins is contained in Supplemental file S5.

2) Describe what some acronyms stand for (e.g. 8-oxo-dGTP).

AU RESPONSE: We have defined this and other acronyms throughout the text as suggested.

3) Are there increased levels of lyso-phospholipids that would confirm AA mobilization by cPLA2?

AU RESPONSE: We thank the reviewer for this comment. The levels of lyso-phospholipids were not able to be detected in the non-targeted metabolomics assay as lyso-phospholipid values are not included in the local metabolomics database.

REVIEWERS' COMMENTS

Reviewer #1 (Remarks to the Author):

NCOMMS-23-04119A

The authors appropriately responded to most of my requests, however, I found some are still needed to be improved, as listed below:

- 1) All "8-OHdG" in Supplementary figures (S7, S8, S15, S16) should be changed to "8-oxo-dG".
- 2) In the abstract, line 33: "MTH1's expression and functional roles in platelets are not known" should be "Expression and functional roles of MTH1 in platelets are not known".
- 3) In discussion, it would be worth to discuss how oxidized nucleotides such as 8-oxo-dGTP or 8-oxoGTP accumulated in MTH1-deficient platelets alter GPCR-dependent platelet function. For example, do such oxidized nucleotides directly affect GPCR signaling or indirectly alter the GPCR signaling through mitochondrial DNA damage?

Reviewer #2 (Remarks to the Author):

The authors performed a substantial amount of additional work to answer detailed, for which they can be complimented. The additional (suppl.) figures are valuable and make the case more clear. One specific comment is that the normalization of platelet responses upon higher concentrations of thrombin is somehow hidden in Figure S4-6, although this is well described on page 6.

These new results also shed doubt on the authors' conclusions regarding the mechanism explaining the phenotype of the KO mice. On page 6 it is stated that "we detected MTH1 expression in the mitochondria of both human and mouse platelets and found that its deficiency in platelets selectively inhibits GPCR-dependent platelet function, mitochondrial ATP production, hemostasis and thrombus formation." The reasoning behind seems to be that thrombin causes mitochondrial ROS production,

which then impairs platelet activation with consequences for thrombosis and hemostasis. However, this cannot be true, as higher thrombin then would lead to a higher extent of platelet dysfunction.

Accordingly, the authors should be more realistic on the interpretation of their data in abstract, results and discussion sections. I see three non-exclusive explanations: (1) the reasoning above, but why then normalization at high thrombin or U46619 doses? (2) The marked alterations in protein phosphorylation levels and metabolites in the KO platelets point to different expression profiles of signaling and metabolic proteins due to an altered ROS status, possibly on the level of MGKs. (3) The altered protein signaling response of the KO platelets agree with the modified thrombosis/hemostasis phenotype and only coincide with the reported mitochondrial changes.

Reviewer #3 (Remarks to the Author):

This is a second review of a manuscript titled "MTH1 protects platelet mitochondria from oxidative damage and is essential for GPCR-dependent platelet function and thrombosis" authored by Ding et al.

Some of my previous comments and questions have been addressed however, there are still some aspects that are not clear.

Major comments:

The level of identification for the metabolites is still not reported in the tables. The authors mentioned that they have an in-house library and that identified metabolites with a score higher than 7 were marked as positively identified. They also mentioned that they used the MSI level system and had levels 0-2. To the best of my knowledge, the MSI level system goes from 1-5, 1 being the highest with MS1, MS/MS and retention time matching with standards. What do they call level 0? Can they provide the citation? In the tables, each metabolite should have the MSI level of identification and not be generalized. It is unclear which metabolites are actually in their in-house library. I couldn't find any information online about their library. Also, <25 ppm mass error is too high for putative identifications, <10 ppm is commonly used. What spectral scoring algorithm was used?

Furthermore, related to metabolite identification and the library the authors replied regarding lyso-phospholipids in their data: "We thank the reviewer for this comment. The levels of lyso-phospholipids were not able to be detected in the non-targeted metabolomics assay as lyso-phospholipid values are not included in the local metabolomics database". This is confusing as lyso-phospholipids were reported

in their tables in the first as well in this revised version of their manuscript. They are listed as LPC, 1-Stearoyl-sn-glycerol 3-phosphocholine (free sn-2 position) or 1-arachidoyl-2-hydroxy-sn-glycero-3-phosphocholine (free sn-2 position), etc.

Interestingly there are several duplicated metabolites (same name but different retention time) such as 1-Stearoyl-sn-glycerol 3-phosphocholine, guanosine monophosphate, inosine monophosphate, etc. How is this possible? Isomers cannot explain +20 seconds retention time difference.

On the same point, the authors replied “we agree with the reviewer that some specialized enrichment and targeted methods are required for the detection the metabolites with retention times less than 1 minutes. However, all of these identified metabolites showed significant differences between two groups based on the OPLS-DA VIP>1 and P value < 0.05”. Not only is the identification of metabolites not retained by the column (below 1 min retention time) but the quantitation is compromised due to ion suppression effects caused by the large number of metabolites present in the ESI source at the same time.

New minor comments:

-page 23 line 658: spectra is already plural, remove “s” in “spectras” .

-same page and line: IDA is not the right acronym, DDA would be the correct one in this case or use information dependent acquisition (IDA).

REVIEWERS' COMMENTS

Reviewer #1 (Remarks to the Author):

NCOMMS-23-04119A

The authors appropriately responded to most of my requests, however, I found some are still needed to be improved, as listed below:

1) All “8-OHdG” in Supplementary figures (S7, S8, S15, S16) should be changed to “8-oxo-dG”.

AU RESPONSE: All “8-OHdG” in Supplementary figures have been changed to “8-oxo-dG”.

2) In the abstract, line 33: “MTH1’s expression and functional roles in platelets are not known” should be “Expression and functional roles of MTH1 in platelets are not known”.

AU RESPONSE: The abstract has been revised as suggested.

3) In discussion, it would be worth to discuss how oxidized nucleotides such as 8-oxo-dGTP or 8-oxoGTP accumulated in MTH1-deficient platelets alter GPCR-dependent platelet function. For example, do such oxidized nucleotides directly affect GPCR signaling or indirectly alter the GPCR signaling through mitochondrial DNA damage?

AU RESPONSE: We thank the reviewer for this comment. Based on the data from the present study, the accumulated oxidized nucleotides resulting from MTH1 deficiency can cause mitochondrial DNA damage, leading to impaired mitochondrial protein synthesis and mitochondrial function. However, we also found an altered cytoplasmic oxidative stress indicated by the impaired ROS-sensitive signaling pathways (p38 and AKT), which might also contribute to the impaired GPCR-dependent signaling in MTH1-deficient platelets apart from the mitochondrial DNA damage.

To clarify this point, the following information has been included in the text on page 16, as follows:

“In addition, our data also imply that altered cytoplasmic oxidative stress indicated by impaired ROS-sensitive signaling pathways (p38 and AKT) might also directly contribute to the observed phenotype (impaired GPCR-dependent signaling) of MTH1-deficient platelets separate from the mitochondrial-related events.”

Reviewer #2 (Remarks to the Author):

1) The authors performed a substantial amount of additional work to answer detailed, for which they can be complimented. The additional (suppl.) figures are valuable and make the case more clear. One specific comment is that the normalization of platelet responses upon higher concentrations of thrombin is somewhat hidden in Figure S4-6, although this is well described on page 6.

AU RESPONSE: We thank the reviewer for the positive comments. The data on the platelet responses to higher concentration of thrombin have been provided in Supplemental Figure S5.

2) These new results also shed doubt on the authors' conclusions regarding the mechanism explaining the phenotype of the KO mice. On page 6 it is stated that "we detected MTH1 expression in the mitochondria of both human and mouse platelets and found that its deficiency in platelets selectively inhibits GPCR-dependent platelet function, mitochondrial ATP production, hemostasis and thrombus formation." The reasoning behind seems to be that thrombin causes mitochondrial ROS production, which then impairs platelet activation with consequences for thrombosis and hemostasis. However, this cannot be true, as higher thrombin then would lead to a higher extent of platelet dysfunction.

AU RESPONSE: We thank the reviewer for this comment and agree that higher doses of thrombin might stimulate more mitochondrial ROS production, potentially leading to a higher degree of platelet dysfunction. However, our study showed a significantly reduced mitochondrial ROS in MTH1-deficient platelets compared to control platelets after thrombin stimulation (Figure 3b), suggesting that the oxidative damage to mtDNA is mainly attributed to MTH1 deficiency and the resultant incorporation of oxidized nucleotides rather than accentuated ROS production.

Higher doses of thrombin or treatment with U49919 normalized the aggregation of MTH1-deficient platelets, likely due to activation of additional signaling pathways, which can bypass the mitochondrial-associated signaling and thereby overcome the platelet dysfunction associated with mitochondrial damage.

3) Accordingly, the authors should be more realistic on the interpretation of their data in abstract, results and discussion sections. I see three non-exclusive explanations: (1) the reasoning above, but why then normalization at high thrombin or U46619 doses? (2) The marked alterations in protein phosphorylation levels and metabolites in the KO platelets point to different expression profiles of signaling and metabolic proteins due to an altered ROS stated, possibly on the level of MGKs. (3) The altered protein signaling response of the KO platelets agree with the modified thrombosis/hemostasis phenotype and only coincide with the reported mitochondrial changes.

AU RESPONSE: We thank the reviewer for this comment. Different doses of thrombin might have distinct impact on platelet activation and function via engagement of different signaling pathways to varying extents since high doses of thrombin have been

shown to be capable of normalizing the impaired aggregation of JNK1^{-/-} (Adam et al., *Blood* 2010), Dab2^{-/-} (Tsai et al., *Arterioscler Thromb Vasc Biol* 2014) or RIP3^{-/-} (Zhang et al., *PNAS* 2017) platelets induced by low dose of thrombin. Therefore, compared to the low dose, high dose of thrombin might stimulate the activation of other signaling pathways, which can bypass the mitochondrial-associated signaling and thereby overcome the platelet dysfunction associated with mitochondrial damage.

To clarify this, the following information has been added in the text on page 13 as follows:

“Importantly, high concentration of GPCR agonists, thrombin or U46619, normalized the impaired aggregation of MTH1-deficient platelets, indicating that the high dose of agonists might stimulate the activation of alternative signaling pathways, which can bypass the mitochondrial-associated signaling and thereby overcome the platelet dysfunction associated with mitochondrial damage.”

Indeed, as the reviewer suggested, the altered platelet function may emanate from gene-associated changes within the parental megakaryocytes resulting in changes to the activation propensity of signaling pathways within MTH1-deficient platelets. We now note this in the text on p13.

“Future studies will delineate the contribution of MTH1 deficiency in megakaryocytes, for example to assess differences in signaling protein expression which may also contribute to the observed platelet functional outcomes.”

Regarding the relationship of altered protein signaling of KO platelets with mitochondrial changes, we agree with the reviewer that they might not be directly linked and this has been stated in the revised manuscript in the text on page 15 as follows:

“However, whilst the phosphorylation changes within proteins such as Slirp and the changes in mitochondrial protein MT-CO1 were coincidental, further work will be required to demonstrate that these two events are functionally linked.”

Reviewer #3 (Remarks to the Author):

This is a second review of a manuscript titled “MTH1 protects platelet mitochondria from oxidative damage and is essential for GPCR-dependent platelet function and thrombosis” authored by Ding et al.

Some of my previous comments and questions have been addressed however, there are still some aspects that are not clear.

Major comments:

The level of identification for the metabolites is still not reported in the tables. The authors mentioned that they have an in-house library and that identified metabolites with a score higher than 7 were marked as positively identified. They also mentioned that they used the MSI level system and had levels 0-2. To the best of my knowledge, the MSI level system goes from 1-5, 1 being the highest with MS1, MS/MS and retention time matching with standards. What do they call level 0? Can they provide the citation? In the tables, each metabolite should have the MSI level of identification and not be generalized. It is unclear which metabolites are actually in their in-house library. I couldn't find any information online about their library. Also, <25 ppm mass error is too high for putative identifications, <10 ppm is commonly used. What spectral scoring algorithm was used?

Furthermore, related to metabolite identification and the library the authors replied regarding lyso-phospholipids in their data: "We thank the reviewer for this comment. The levels of lyso-phospholipids were not able to be detected in the non-targeted metabolomics assay as lyso-phospholipid values are not included in the local metabolomics database". This is confusing as lyso-phospholipids were reported in their tables in the first as well in this revised version of their manuscript. They are listed as LPC, 1-Stearoyl-sn-glycerol 3-phosphocholine (free sn-2 position) or 1-arachidoyl-2-hydroxy-sn-glycero-3-phosphocholine (free sn-2 position), etc.

Interestingly there are several duplicated metabolites (same name but different retention time) such as 1-Stearoyl-sn-glycerol 3-phosphocholine, guanosine monophosphate, inosine monophosphate, etc. How is this possible? Isomers cannot explain +20 seconds retention time difference.

On the same point, the authors replied "we agree with the reviewer that some specialized enrichment and targeted methods are required for the detection the metabolites with retention times less than 1 minutes. However, all of these identified metabolites showed significant differences between two groups based on the OPLS-DA VIP>1 and P value < 0.05". Not only is the identification of metabolites not retained by the column (below 1 min retention time) but the quantitation is compromised due to ion suppression effects caused by the large number of metabolites present in the ESI source at the same time.

New minor comments:

-page 23 line 658: spectra is already plural, remove "s" in "spectras".

-same page and line: IDA is not the right acronym, DDA would be the correct one in this case or use information dependent acquisition (IDA).

AU RESPONSE: We thank the reviewer for these comments. Due to technical and other issues, and the dispensable role of these data for the overall conclusions drawn from this study, after further discussion amongst the author group, we elected to remove the metabolomics data from the manuscript. This will allow us to properly experimentally address the concerns raised by the reviewer, with the goal for the resultant data to form part of a new publication.